# TERRA regulate the transcriptional landscape of pluripotent cells through TRF1-dependent recruitment of PRC2

Rosa María Marión[1†], Juan J Montero[1†], Isabel López de Silanes[1],
Osvaldo Graña-Castro[2], Paula Martínez[1], Stefan Schoeftner[1‡§],
José Alejandro Palacios-Fábrega[1#], Maria A Blasco[1*]

[1]Telomeres and Telomerase Group, Molecular Oncology Program, Spanish National Cancer Centre (CNIO), Madrid, Spain; [2]Bioinformatics Unit, Structural Biology Program, Spanish National Cancer Centre (CNIO), Madrid, Spain

**\*For correspondence:**
mblasco@cnio.es

[†]These authors contributed equally to this work

**Present address:** [‡]Genomic Stability Unit, Laboratorio Nazionale del Consorzio Interuniversitario per le Biotecnologie (LNCIB), Trieste, Italy; [§]Department of Life Sciences, Università degli Studi di Trieste, Trieste, Italy; [#]Astellas Pharma Europe Ltd, Chertsey, United Kingdom

**Abstract** The mechanisms that regulate pluripotency are still largely unknown. Here, we show that Telomere Repeat Binding Factor 1 (TRF1), a component of the shelterin complex, regulates the genome-wide binding of polycomb and polycomb H3K27me3 repressive marks to pluripotency genes, thereby exerting vast epigenetic changes that contribute to the maintenance of mouse ES cells in a naïve state. We further show that TRF1 mediates these effects by regulating TERRA, the lncRNAs transcribed from telomeres. We find that TERRAs are enriched at polycomb and stem cell genes in pluripotent cells and that TRF1 abrogation results in increased TERRA levels and in higher TERRA binding to those genes, coincidental with the induction of cell-fate programs and the loss of the naïve state. These results are consistent with a model in which TRF1-dependent changes in TERRA levels modulate polycomb recruitment to pluripotency and differentiation genes. These unprecedented findings explain why TRF1 is essential for the induction and maintenance of pluripotency.

DOI: https://doi.org/10.7554/eLife.44656.001

## Introduction

Multiple cellular processes, including pluripotency and the determination of cell fate, are regulated by epigenetic modifications, including genome-wide changes in DNA and histone methylation (*Blasco, 2007*; *Theunissen and Jaenisch, 2014*). One key histone modification, tri-methylation of lysine 27 in histone 3 (H3K27me3), is a transcription repression mark that it is controlled by the polycomb transcriptional repressor (PRC) proteins (*Sparmann and van Lohuizen, 2006*). There are two polycomb complexes, PRC1 and PRC2, which cooperate to achieve gene silencing (*Lund and van Lohuizen, 2004*). PRC2 encompasses EED (Embryonic ectoderm development), EZH2 (Enhancer of zeste), SUZ12 (Suppressor of zeste 12), and ESC (Extra sex combs) (*Ringrose and Paro, 2004*), and is involved in the initiation of gene repression. In particular, the PRC2 component EZH2 is a SET-domain-containing protein that catalyzes the repressive H3K27me3 mark and to lesser extent also H3K9me3 (*Simon and Kingston, 2009*). PRC2-mediated H3K27me3 recruits PRC1, which contributes to gene silencing, possibly by blocking transcriptional elongation by the RNA Polymerase II (*Stock et al., 2007*).

Interestingly, polycomb PRC1 and PRC2 complexes have been proposed to have a key role in the maintenance of embryonic stem (ES) cell pluripotency (*Pereira et al., 2010*). In particular, polycomb proteins are important for restraining the activity of lineage-specifying factors in ES cells, and also have been shown to be essential in establishing the conversion of differentiated cells towards pluripotency (*Azuara et al., 2006*; *Boyer et al., 2006*; *Endoh et al., 2008*). PRC2 proteins and the

H3K27me3 mark are proposed to maintain genes in a state in which they are poised for transcription and contribute to pluripotency (*Azuara et al., 2006*; *Boyer et al., 2006*; *Endoh et al., 2008*). In agreement with this notion, EZH2 is upregulated during reprogramming and EZH2 knock-down impairs reprogramming. Also, PRC2 is essential for pluripotency but is not essential in human ES cells in the 'naïve' state (*Shan et al., 2017*).

In ES cells, a significant proportion of polycomb target genes are repressed genes that encode transcription factors that are required for lineage specification later during development (*Boyer et al., 2006*; *Lee et al., 2006*). These genes are also co-occupied by the key pluripotency factors OCT4, NANOG, and SOX2, suggesting that the function of these pluripotency genes in repressing gene expression may be mediated by polycomb (*Boyer et al., 2005*; *Boyer et al., 2006*). Indeed, downregulation of the PRC1 and PRC2 activity in ES cells leads to global de-repression of these genes and to unscheduled differentiation (*Boyer et al., 2006*; *Endoh et al., 2008*). Deletion of one of the PRC2 components in mice (EED, SUZ12 or EZH2) results in severe defects in development, suggesting the mis-expression of lineage-specific genes (*Faust et al., 1995*; *O'Carroll et al., 2001*; *Pasini et al., 2004*). However, the deletion of PRC2 components in ES cells is not lethal (*Leeb et al., 2010*).

Interestingly, polycomb has also been reported to bind to bivalent genes, which are occupied by both the heterochromatic mark H3K27me3 and the active mark H3K4me3. In this scenario, PRC2 is important in maintaining lineage genes in a poised state ready to respond to differentiation cues (*Voigt et al., 2013*).

Telomeres are special heterochromatin structures at chromosome ends, which are formed by tandem repeats of the TTAGGG sequence bound by the so-called shelterin complex (*Blackburn, 2005*; *de Lange, 2005*; *Martínez and Blasco, 2011*). Telomeric chromatin is enriched in histone-repressive marks including H3K9me3 and H4K20me3 (*García-Cao et al., 2002*; *García-Cao et al., 2004*; *Gonzalo et al., 2005*; *Gonzalo et al., 2006*; *Benetti et al., 2007*; *Blasco, 2007*). Recently, we also reported that telomeres are enriched for the PRC2-repressive mark H3K27me3 (*Montero et al., 2018*). The function of telomeres is to protect chromosome ends from degradation and from triggering chromosomal aberrations, such as chromosome end-to-end fusions. Two independent studies showed that the shelterin component TRF1 is also greatly upregulated during reprogramming (*Boué et al., 2010*; *Schneider et al., 2013*). Indeed, we showed that TRF1 upregulation is an early event during cellular reprogramming, which precedes and is independent of telomere elongation by telomerase (*Schneider et al., 2013*). The *Terf1* gene is a direct target of OCT4, and is also essential for the induction and maintenance of pluripotency. In support of this, deletion of TRF1 causes embryonic lethality at the blastocyst stage (*Karlseder et al., 2003*). More recently, we showed that TRF1 is also upregulated during in vivo reprogramming, showing a similar pattern of expression to that of OCT4 in reprogrammed tissues (*Marión et al., 2017*). In spite of this solid evidence that TRF1 has an important role in pluripotency, the mechanisms that allow TRF1 to perform this mediating role have remained unknown until now.

PRC2 can interact both in vivo and in vitro with the long non-coding RNAs transcribed from telomeres, or TERRA, and this interaction is essential for the establishment of the H3K27me3 mark at telomeres (*Chu et al., 2017*; *Wang et al., 2017*; *Montero et al., 2018*). TERRA has also been shown to be associated with polycomb marks in the vicinity of genes and to modulate gene expression (*Chu et al., 2017*). Thus, there seems to be an interplay between telomere transcriptional status and long-range epigenetic regulation. In fact, PRC2 interacts with several long non-coding RNAs (lncRNAs), and this interaction is thought to regulate gene expression by recruiting PRC2 to specific loci. Some examples of lncRNAs that can physically interact with PRC2 and recruit it to specific loci include *Xist* (*Zhao et al., 2008*), *Hotair* (*Rinn et al., 2007*) and the antisense non-coding RNA in the *Cdkn2a* locus (*Yap et al., 2010*). These lncRNAs play important roles in X chromosome activation and tumorigenesis. However, how a lncRNA is able to provide specificity for PRC2 recruitment is not clear.

In addition, TERRA has been previously described to interact with the shelterin component TRF2, which can interact with TRF1, thus opening the possibility that polycomb may also be interacting with shelterin components. In this regard, a recent report showed that the *Arabidopsis thaliana* telomere-repeat binding factors (TRBs) recruit PRC proteins to different promoters through a telobox motif. In the absence of the three TRB proteins, the PRC2-mediated H3K27me3 mark was altered in

a similar manner to that of PRC2 mutants. Indeed, an interaction between TRB1–3 and PRC2 proteins was found (*Zhou et al., 2016b*; *Zhou et al., 2018*).

Here, we set to address the mechanisms through which OCT4-mediated TRF1 upregulation functions as an essential process for the induction and maintenance of pluripotency in mouse cells. To this end, we have used an unbiased genome-wide approach, looking for global changes in gene expression in the absence of TRF1. We make the unprecedented finding that TRF1 abrogation has a 'butterfly effect' on the transcription of naïve pluripotent cells, altering the epigenetic landscape of these cells through a novel mechanism, which involves TERRA-mediated polycomb recruitment to pluripotency genes and cell-fate genes.

## Results

### Abrogation of TRF1 in 2i-grown iPS cells changes the expression of genes related to pluripotency, differentiation and control by polycomb

To address whether TRF1 abrogation results in genome-wide changes in gene expression that could explain why TRF1 is required for pluripotency, we set to analyze the whole cellular transcriptome directly in induced pluripotent stem cells (iPS) cells in which TRF1 had been severely downregulated by the use of a short hairpin RNA (shRNA) (*Figure 1A*). We used *Trp53* (also known as p53)-null iPS cells to allow for cell proliferation in the absence of TRF1 (*Martínez et al., 2009*). We optimized the experiment so that the cells were harvested as soon as abrogation of TRF1 could be confirmed, in order to avoid deleterious effects associated to TRF1 depletion, such as increased DNA damage response. As expected from TRF1 downregulation, we observed an increase in the number of multitelomeric signals (MTS) (*Figure 1—figure supplement 1A*), which have previously been related to TRF1 abrogation (*Martínez et al., 2009*; *Sfeir et al., 2009*). In these conditions, we did not find a significant increase in the DNA damage marker γH2AX in TRF1-depleted cells compared to control cells using either western blot (*Figure 1—figure supplement 1B*) or immunofluorescence against γH2AX (*Figure 1—figure supplement 1C*). Similarly, we did not find any significant difference in DNA damage specifically located at telomeres, as determined by performing double immunofluorescence against γH2AX and the telomeric protein RAP1 together with quantification of the number of colocalizations or Telomere-Induced Foci (TIFs) (*Figure 1—figure supplement 1D*). These results indicate that the short period of cell harvesting after TRF1 depletion is not sufficient to allow the activation of a DNA damage response (DDR) that is greater than the basal levels shown by control cells, thus ruling out the possibility that the transcriptome effects observed here may be due to increased DNA damage resulting from TRF1 depletion. To determine the global gene expression changes that are associated with TRF1 depletion, we next performed RNA sequencing (RNA-seq) (see Materials and methods) in control *Trp53*-null 2i-grown iPS cells and *Trp53*-null 2i-grown iPS cells with downregulated TRF1 levels (*Figure 1A*). Gene set enrichment analysis (GSEA) revealed that genes that are downregulated when TRF1 is deleted, such as MYC, SOX2, NANOG and BMP (*Figure 1B–J*), are overrepresented in gene sets that are targets of pluripotency factors and pluripotency pathways. Interestingly, we found a clear enrichment in genes regulated by MYC (*Figure 1B*), concomitant with the fact that *Myc* was indeed downregulated by 3.8-fold in the absence of TRF1 in this RNA-seq experiment (Table S1). Also, genes that are downregulated when TRF1 is depleted were enriched among the gene targets of SOX2 and NANOG (*Figure 1C,D*), in genes that are upregulated in ES cells (*Figure 1E*) and in genes that are downregulated during the differentiation of embryoid bodies (*Figure 1F*). Together, these observations suggest that depletion of TRF1 induces the loss of pluripotency and initiates the differentiation of the iPS cells. In agreement with this notion, genes that are upregulated in the absence of TRF1 were enriched in gene sets that are upregulated during the differentiation of embryoid bodies from ES cells (*Figure 1H*), in targets of SUZ12 (*Figure 1I*), and in genes that are upregulated upon *knockout* of *Bmp2* (*Figure 1J*), a member of bone morphogenetic proteins that are important in tissue differentiation (*Zhou et al., 2016a*).

Further analysis of the RNA-seq results showed that 328 genes were downregulated and 483 genes upregulated significantly in the absence of TRF1 (*Figure 1K*). Analysis of the genes that are downregulated when abrogating TRF1 by Enrichr (a comprehensive gene set enrichment analysis web server) showed a significant enrichment in the targets of a large number of pluripotency

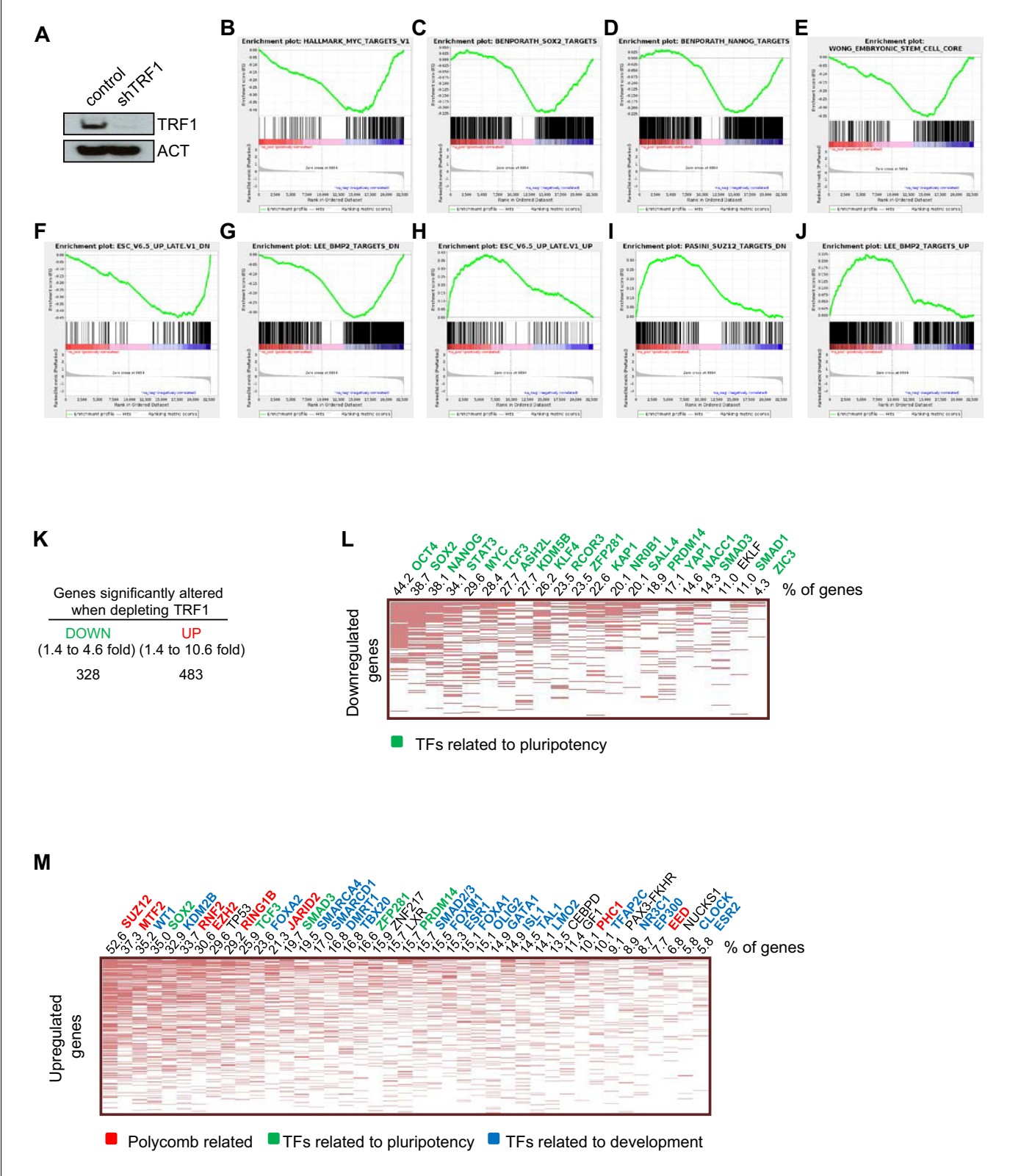

**Figure 1.** Transcriptome of TRF1-depleted iPS cells. (**A**) Western blot of TRF1 protein and actin (ACT) as the loading control in control and TRF1-depleted iPS cells. (**B–J**) Gene expression data obtained by RNA-seq of two independent experiments of TRF1-depletion in iPS cells, analyzed by Gene Set Enrichment Analysis (GSEA) to determine significantly enriched gene sets. Highly significant gene sets are shown here. Genes that were downregulated upon TRF1 deletion were enriched in targets of MYC (**B**), SOX2 (**C**), NANOG (**D**) and BMP2 (**G**). They were also enriched in

*Figure 1 continued on next page*

*Figure 1 continued*

genes that are expressed in ES cells (E) and in genes that are downregulated during differentiation (F). Genes that are upregulated upon TRF1 deletion were enriched in genes that are also upregulated during differentiation (H) and in targets of SUZ12 (I) and BMP2 (J). (K) Changes of gene expression in TRF1-depleted iPS cells analyzed by RNA-seq. The table summarizes the numbers of expressed transcripts that are expressed differentially in TRF1-depleted and control iPS cells. (L) Analysis using Enrichr of genes that are downregulated in TRF1-deleted iPS cells relative to control cells. The clustergram, representing the results of a CHEA analysis, shows that an important number of downregulated genes are targets of numerous pluripotency factors. The numbers on the top of the clustergram indicate the percentage of all of the downregulated genes that are targets of each pluripotency factor. Note that more than 40% of all downregulated genes are bound by OCT4. (M) Analysis using Enrichr of genes that are upregulated in TRF1-deleted iPS cells relative to control cells. The clustergram, representing the results of a CHEA analysis, shows that an important number of upregulated genes are mainly targets of polycomb-related proteins (red) and differentiation-related proteins (blue). The numbers on the top of the clustergram indicate the percentage of all of the upregulated genes that are targets of each factor.

DOI: https://doi.org/10.7554/eLife.44656.002

The following figure supplements are available for figure 1:

**Figure supplement 1.** TRF1 abrogation in 2i-grown iPS cells does not increase DNA damage.

DOI: https://doi.org/10.7554/eLife.44656.003

**Figure supplement 2.** Functional annotation of genes altered when depleting TRF1.

DOI: https://doi.org/10.7554/eLife.44656.004

factors, including NANOG, OCT4, KLF4, SOX2 and MYC, with *Myc* again being one of the most downregulated genes after depletion of TRF1 (*Figure 1L*). *Figure 1L* shows the percentages of the downregulated genes that are direct targets of the indicated transcription factors. Strikingly, up to 86% of the downregulated genes were targets of pluripotency factors. In particular, more than 40% of all downregulated genes were direct targets of OCT4 and more than 30% were direct targets of SOX2, NANOG and STAT3 (*Figure 1L*). These findings indicate that TRF1 depletion alters the expression of numerous genes that are controlled by key pluripotency transcription factors. In line with these findings, Enrichr analysis of the downregulated genes show that they were significantly enriched in genes encoding components of signaling pathways regulating the pluripotency of stem cells; 26 of these genes belong to PluriNetWork, which contains genes underlying pluripotency in mouse (*Figure 1—figure supplement 2A*). All together, these results indicate that TRF1 depletion in 2i-grown iPS cells reduces the expression of genes that are implicated in maintaining pluripotency programs.

Interestingly, similar Enrichr analysis of genes that are upregulated when depleting TRF1 (*Figure 1M*) showed a significant enrichment in the targets of the polycomb repressive complexes PRC1 and PRC2, together with pluripotency factors and factors related to development (*Figure 1M*). In particular, up to 82% were targets of differentiation factors, 64% were targets of PRC2, and 65% were targets of pluripotency factors. Wikipathways and KEGG pathways analysis revealed that the upregulated genes were also enriched in important pathways regulating pluripotency and differentiation, such as the BMP, Tgfβ and WNT signaling pathways (*Figure 1—figure supplement 2B* left panel). In agreement with the unexpected finding that PRC2 targets were greatly enriched in TRF1-depleted cells, we found that the upregulated genes were enriched in the polycomb-related chromatin marks H3K27me3 and H3K9me3, which are characteristic of heterochromatin (*Figure 1—figure supplement 2B* right panel). We also found that thesegenes are enriched in H3K4me3, a mark present in bivalent genes, where polycomb has been described to bind in primed ES cells (*Voigt et al., 2013*) (*Figure 1—figure supplement 2B* right panel). These findings suggest that TRF1 depletion in 2i-grown iPS cell results in the re-activation of polycomb target genes and genes containing the polycomb repressive mark H3K27me3, a phenomenon associated with the loss of the pluripotency stage and the expression of lineage genes. Thus, pathways that are important in lineage commitment and pluripotency, such as the BMP, Tgfβ and WNT signaling pathways, are altered, consistent with loss of the pluripotency state. In summary, these findings indicate that abrogation of TRF1 expression in 2i-grown iPS cells alters the expression of genes that are related to pluripotency, differentiation, and polycomb complexes, supporting the hypothesis that the loss of TRF1 induces the loss of pluripotency and the induction of differentiation.

## Abrogation of TRF1 in 2i-grown iPS cells alters global SUZ12 and H3K27me3 genomic distribution

As TRF1 deletion alters the expression of polycomb- and H3K27me3-regulated genes, we next set out to address whether TRF1 affects the global genome distribution of SUZ12 and H3K27me3 proteins. To do so, we performed chromatin immunoprecipitation of SUZ12 and H3K27me3 followed by deep sequencing (ChIP-seq) in control 2i-grown iPS cells and 2i-grown iPS cells in which TRF1 had been severely downregulated (*Figure 1A*). Interestingly, we found that TRF1 abrogation significantly increased the number of SUZ12 binding sites within the genome (4230 peaks) compared to that in control cells (440 peaks) (*Figure 2A*, and *Figure 2—figure supplement 1*). Heat maps of the read distribution around the SUZ12 peaks confirmed a clear increase in SUZ12 deposition in these genomic regions upon TRF1 depletion, indicating that abrogation of TRF1 in 2i-grown iPS induces a strong recruitment of SUZ12 into the genome (*Figure 2B*). Total levels of SUZ12 protein were not altered upon TRF1 depletion in four independent experiments (*Figure 2C*, see bottom panel for quantification), indicating that the recruitment of SUZ12 to the genome is not due to overexpression of the SUZ12 protein. Then, we used Enrichr to analyze the characteristics of the genes that were associated with the genome sites where we observed an increased binding of SUZ12 protein upon depletion of TRF1 (see Clusters 1, 2 and 3 of the SUZ12 heatmap [*Figure 2B*]). Importantly, CHEA transcription factor analysis showed that the genes that were associated with sites of SUZ12 recruitment upon TRF1 deletion are the targets of a great number of pluripotency factors, including OCT4, NANOG, KLF4 and MYC (Table S2), as well as of components of the polycomb repressor complexes (Table S2). In addition, KEGG AND Wikipathway analysis showed that these genes are enriched in several pathways that are involved in the regulation of ESC pluripotency, such as the WNT and TGFβ pathways, pathways related to differentiation, and pathways related to the nervous system (*Figure 2D*, Table S3). Furthermore, analysis of GO Biological process indicated that these genes are highly involved in processes related to the nervous system (*Figure 2—figure supplement 2*). We performed a second ChIP-seq to confirm our results, and we found that the heatmaps of the two biological replicates of SUZ12 ChIP-seq, with reads plotted around the peaks for both replicates, show the same pattern (*Figure 2—figure supplement 1A*). Importantly, the analysis of the genomic regions in which the SUZ12 signal was increased was very similar (*Figure 2—figure supplement 1A*). Thus, we observed that all of the clusters of these heatmaps included regions where SUZ12 increases upon TRF1 depletion in both replicas (*Figure 2—figure supplement 1A*). Moreover, we showed that 99% of the genes that showed increased levels of SUZ12 upon TRF1 depletion in the first experiment also showed higher levels of SUZ12 in the second replicate, confirming the close similarity between both ChIP-seq replicates and the recruitment of SUZ12 to the described genomic sites (*Figure 2—figure supplement 1C*). These results suggest that TRF1 may control the expression of pluripotency- and differentiation-related proteins by directly or indirectly inducing the redistribution of the SUZ12 PRC2 component throughout the genome.

To further support our observation from ChIP-seq that TRF1 depletion increases PRC2 recruitment to the genome, we next performed an additional ChIP-seq analysis to study the H3K27me3 polycomb mark upon TRF1 abrogation. We found 14,317 H3K27me3 peaks in control 2i-grown iPS cells and 16,165 H3K27me3 peaks in TRF1-depleted 2i-grown iPS cells (*Figure 2E* and *Figure 2—figure supplement 1*). Heat maps of the read distribution around H3K27me3 peaks (*Figure 2F*) showed a cluster of peaks (Cluster 1) in which a clear increase of H3K27me3 deposition was observed. Total levels of H3K27me3 were not altered upon TRF1 depletion in four independent experiments (*Figure 2G*, see quantification in bottom panel), indicating that the deposition of H3K27me3 to those sites in the genome is not due to changes in the cellular levels of H3K27me3. We then use Enrichr to analyze the characteristics of the genes associated with the genome sites where we observed increased binding of H3K27me3 protein upon depletion of TRF1 (Cluster 1 of the H3K27me3 heatmap from *Figure 2F*). KEGG and Wikipathway analyses showed results that were very similar to those of SUZ12, with a clear enrichment in pathways regulating pluripotency, differentiation, and the nervous system (*Figure 2H*). We performed a second ChIP-seq to confirm our results, and found that the heatmaps of the two biological replicates of H3K27me3 ChIP-seq, with reads plotted around the peaks for both replicates, show the same pattern (*Figure 2—figure supplement 1D*). We observed that clusters 3 and 7 of these heatmaps include regions where H3K27me3 increases upon TRF1 depletion in both replicates (*Figure 2—figure supplement 1D*).

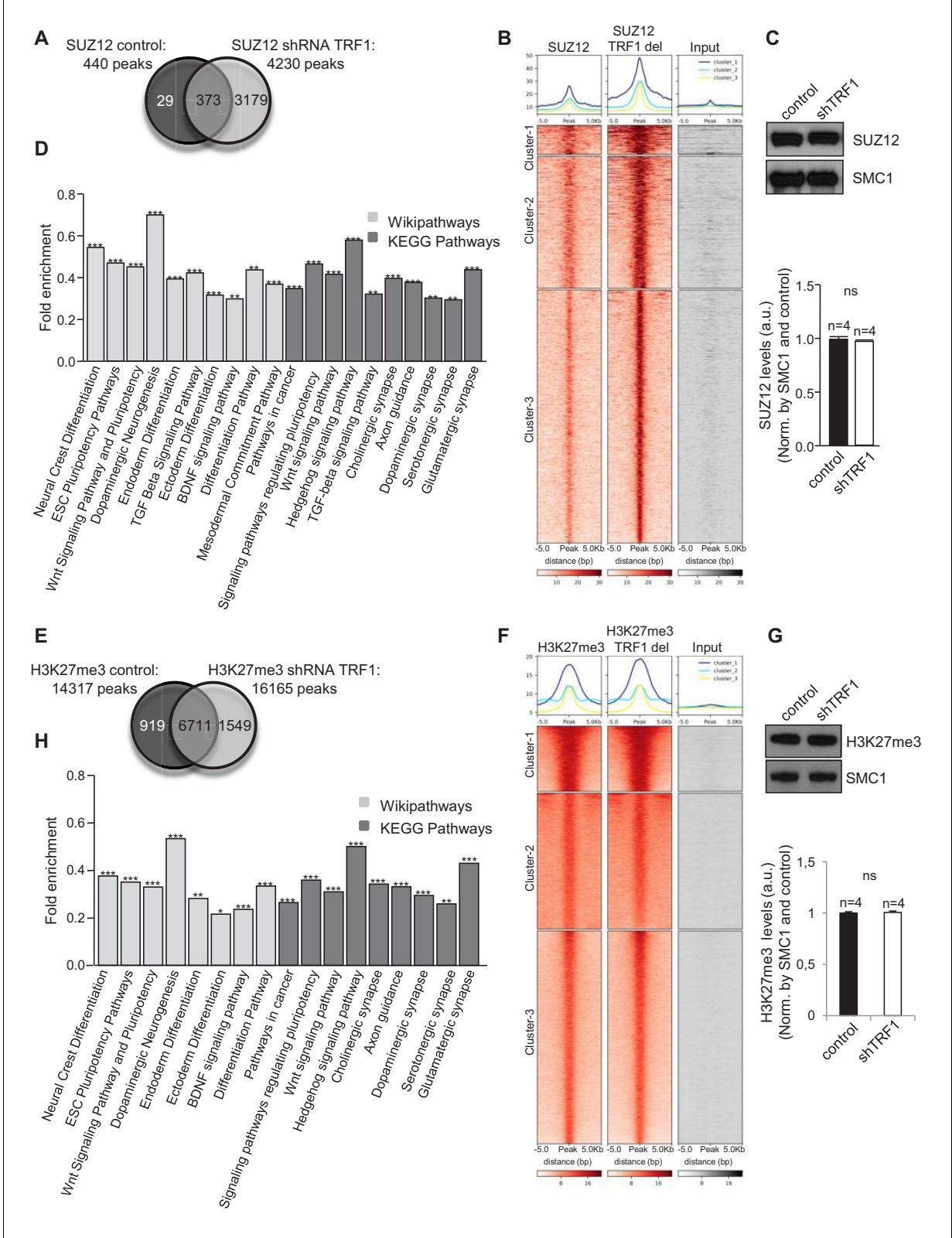

**Figure 2.** Abrogation of TRF1 alters SUZ12 and H3K27me3 genome localization. Analysis of the genome-wide binding of SUZ12 and H3K27me3 by chromatin immunoprecipitation followed by deep sequencing (ChIP-seq) in control and TRF1-depleted 2i-grown iPS cells. (**A**) Number of SUZ12 binding peaks in control and TRF1-depleted cells. Note that TRF1 abrogation significantly increases the number of SUZ12 binding sites in the genome (4230 peaks) compared to that in control cells (440 peaks). The Venn diagram shows the number of genes that are annotated to SUZ12 peaks in both

*Figure 2 continued on next page*

*Figure 2 continued*

conditions. (B) Heat maps of the reads distribution around 5 Kb of SUZ12 peaks in control and TRF1-depleted cells. Note the visible increase in SUZ12 deposition in these sites upon TRF1 depletion. (C) (Top) Representative image of Western blots of SUZ12 and SMC1 (as loading control) from control and TRF1-depleted iPS cells. (Bottom) Quantification of SUZ12 levels in four independent experiments. Note that SUZ12 protein levels are not affected by the depletion of TRF1 protein. n = number of independent experiments. Error bars = SE. Statistical analysis, Student's t-test. (D) KEGG AND Wikipathways analysis of the genes associated with the sites of the genome were increased binding of SUZ12 protein was observed upon abrogation of TRF1. Genes annotated in Clusters 1, 2 and 3 of the SUZ12 heatmap (*Figure 2B*) were used. Note that these genes are mainly enriched in pathways related to pluripotency, differentiation and the development of the nervous system. ** = Adjusted p value <0.01, *** = Adjusted p value <0.001. (E) Number of H3K27me3 binding peaks in control and TRF1-deleted cells. Note that TRF1 abrogation increases the number of H3K27me3 binding sites within the genome (16,165 peaks) compared to that in control cells (14,317 peaks). The Venn diagram shows the number of genes annotated to H3K27me3 peaks in both conditions. (F) Heat maps of the reads distribution around 5 Kb of H3K27me3 peaks in control and TRF1-depleted cells. Note that Cluster 1 shows an increase in H3K27me3 deposition upon TRF1 downregulation. (G) (Top) Representative image of Western blots of H3K27me3 and SMC1 (as loading control) from control and TRF1-depleted iPS cells. (Bottom) Quantification of H3K27me3 levels in four independent experiments. n = number of independent experiments. Error bars = SE. Statistical analysis, Student's t-test. Note that H3K27me3 protein levels are not affected by the depletion of TRF1 protein. (H) KEGG and Wikipathways analysis of the genes associated with the genome sites were increased binding of H3K27me3 protein was observed upon depletion of TRF1. Genes annotated in Cluster 1 of the H3K27me3 heatmap (*Figure 2F*) were used. Note that the results are very similar to those obtained for SUZ12. ** = Adjusted p value <0.01, *** = Adjusted p value <0.001.

DOI: https://doi.org/10.7554/eLife.44656.005

The following figure supplements are available for figure 2:

**Figure supplement 1.** Validation of SUZ12 and H3K27me3 ChIP-seq.
DOI: https://doi.org/10.7554/eLife.44656.006
**Figure supplement 2.** Functional annotation analysis of genes enriched in SUZ12 upon TRF1 depletion.
DOI: https://doi.org/10.7554/eLife.44656.007
**Figure supplement 3.** Abrogation of TRF1 in 2i-grown iPS cells induces the loss of the naïve state and a transition to a primed or differentiated state.
DOI: https://doi.org/10.7554/eLife.44656.008
**Figure supplement 4.** Extratelomeric binding of TRF1 in 2i-grown iPS cells.
DOI: https://doi.org/10.7554/eLife.44656.009

Importantly, we showed that 90% of the genes that showed increased levels of H3K27me3 upon TRF1 depletion in the first experiment also showed higher levels of H3K27me3 in the second replicate, confirming the close similarity between the two ChIP-seq experiments (*Figure 2—figure supplement 1F*). All together, these results indicate that the depletion of TRF1 in 2i-grown iPS cells induces a dramatic recruitment of SUZ12 and H3K27me3 to genes controlling key pathways in pluripotency and differentiation.

## Depletion of TRF1 induces the recruitment of SUZ12 and H3K27me3 to genes that are de-regulated in the absence of TRF1

To further understand how TRF1 influences pluripotency by affecting polycomb distribution, we made a Venn diagram showing genes that are bound by SUZ12 and H3K27me3 specifically in the absence of TRF1 and genes that are downregulated in the absence of TRF1 (*Figure 3A*). We found a set of 14 genes that were downregulated and that also recruited SUZ12 and H3K27me3 upon TRF1 abrogation (*Figure 3A*). Importantly, *Myc* was present within this set of genes, and showed a clear gain of SUZ12 and H3K27me3 peaks in the absence of TRF1 (*Figure 3B*). Of special interest in this set of genes were *Myc*, *Id1* and *Foxd3*, which are involved in control of pluripotency, the TGFβ pathway and the WNT pathway, respectively. These findings indicate that TRF1 abrogation induces the recruitment of SUZ12 and H3K27me3 to key pluripotency and development genes, thereby repressing their expression. We then selected a set of 30 genes that showed recruitment of either SUZ12 or H3K27me3, or of both SUZ12 and H3K27me3 when TRF1 was depleted and validated their expression by q-PCR in three TRF1-downregulation-independent experiments (*Figure 3C*). As control, we confirmed *Terf1* downregulation (*Figure 3C*). Importantly, we confirmed downregulation of all of the genes including *Myc*, thus validating the RNA-seq findings (*Figure 3C*). Next, to obtain a more accurate picture of SUZ12 and H3K27me3 binding to the genes that are downregulated in the absence of TRF1, we created heat maps of the distribution of SUZ12 and H3K27me3 reads within 2.5 Kb of the transcription start sites (TSS) of all of the genes that were significantly downregulated in the RNA-seq upon TRF1 abrogation (*Figure 3D*). We observed a cluster of genes with dramatic recruitment of SUZ12 to their TSS upon depletion of TRF1 (*Figure 3D*, Cluster 1 of SUZ12 heatmap),

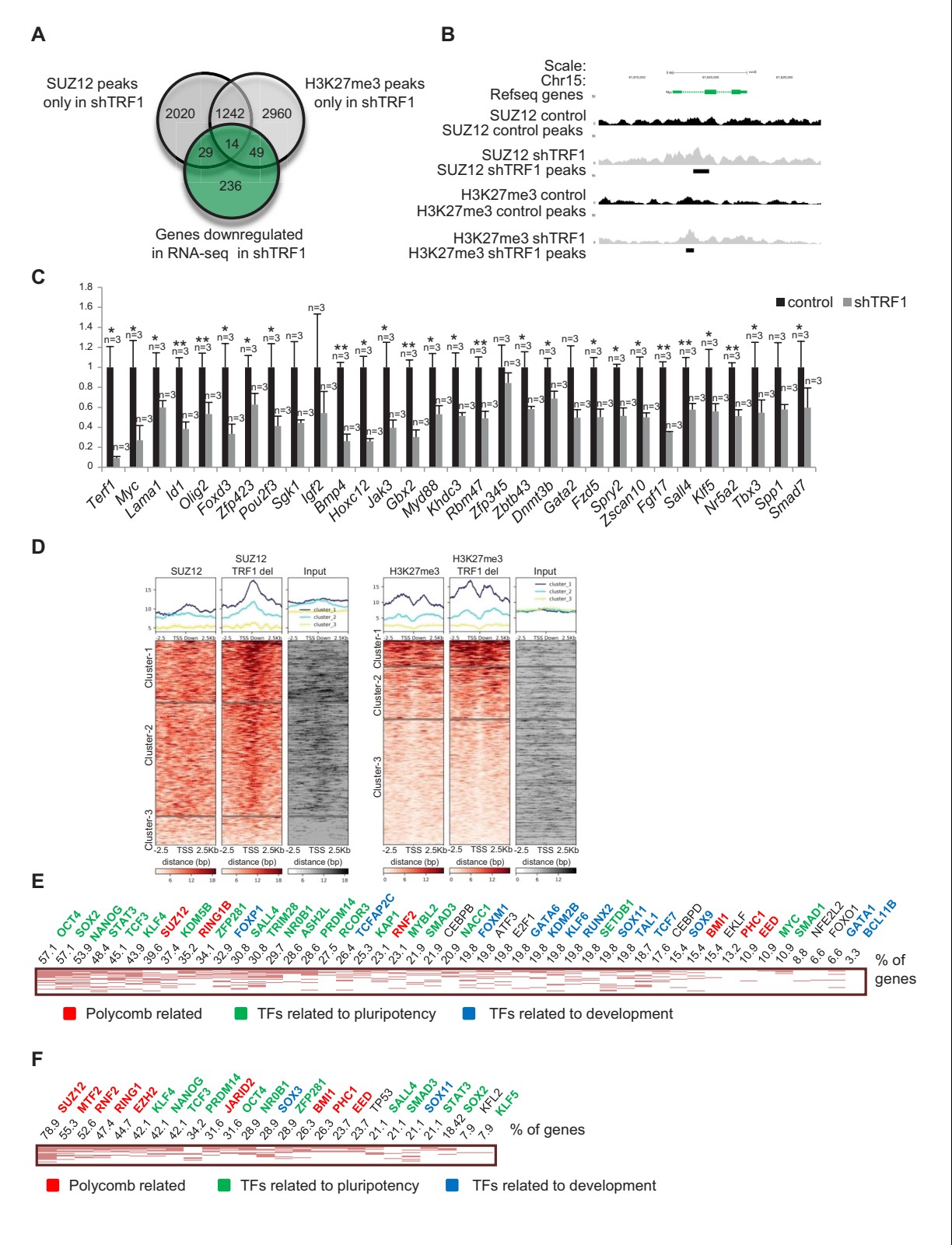

**Figure 3.** Depletion of TRF1 induces the recruitment of SUZ12 and H3K27me3 to genes that are downregulated in the absence of TRF1. (A) Venn diagram showing overlap of genes bound by SUZ12 and H3K27me3 specifically in the absence of TRF1 with the set of genes that are downregulated in this same condition, as obtained by RNA-seq. Note that a number of downregulated genes gain SUZ12 and H3K27me3 peaks upon TRF1 abrogation. (B) Gain of SUZ12 and H3K27me3 peaks in the *Myc* gene. (C) Gene expression analysis by q-PCR of genes that are downregulated upon TRF1

*Figure 3 continued on next page*

*Figure 3 continued*

depletion in which SUZ12 or H3K27me3 recruitment was detected. Note that abrogation of TRF1 induces a clear reduction in the expression of these genes, as observed before by RNA-seq. As a control, we confirmed that TRF1 expression was drastically decreased. n = number of independent experiments. Error bars = SE. Statistical analysis, one tail, paired Student's t-test. * = p value<0.05; ** = p value<0.01. (D) Heat maps of the distribution of reads of SUZ12 and H3K27me3 within 2.5 Kb of the transcription start sites (TSS) of all of the genes that are significantly downregulated in the RNA-seq upon TRF1 abrogation. Note that Clusters 1 of both the SUZ12 and the H3K27me3 heatmap show a clear recruitment of the corresponding protein. (E) Genes belonging to Cluster 1 of the Suz12 heatmap (*Figure 3D*), which present a dramatic enrichment of SUZ12 upon TRF1 depletion, were analyzed. The clustergram , representing the results of the CHEA analysis, shows that an important number of these genes are targets of numerous pluripotency factors (green). The numbers on the top of the clustergram indicate the percentage of all the downregulated genes that are targets of each factor. (F) Genes belonging to Cluster 1 of the H3K27me3 heatmap (*Figure 3D*), which present a clear enrichment of H3K27me3 upon TRF1 depletion, were analyzed. As in the case of SUZ12, the clustergram shows that an important number of these genes are targets of pluripotency factors (green).

DOI: https://doi.org/10.7554/eLife.44656.011

The following figure supplement is available for figure 3:

**Figure supplement 1.** Functional analysis of genes downregulated in the absence of TRF1 that show recruitment of SUZ12 and H3K27me3.

DOI: https://doi.org/10.7554/eLife.44656.012

and a cluster with a more moderate increase in SUZ12 binding (Cluster 2). For H3K27me3, we detected a cluster of genes (Cluster 1) that recruited H3K27me3 to their TSS upon depletion of TRF1 (*Figure 3D*). We then performed an Enrichr analysis of the genes included in Cluster 1 of the SUZ12 heatmaps, corresponding to those genes that present a clearer difference in binding of SUZ12 upon TRF1 depletion. The results showed that these genes were significantly enriched in direct targets of a high number of pluripotency factors, including OCT4, SOX2, NANOG and MYC, among others (*Figure 3E*). As expected, they were also enriched in direct targets of the polycomb complexes. Wikipathways and KEGG analysis showed that these genes were significantly enriched in important pathways controlling pluripotency and differentiation (*Figure 3—figure supplement 1A*). Analysis of the genes that are enriched in H3K27me3 upon depletion of TRF1 (Cluster 1 of H3K27me3 heatmap) showed similar results (*Figure 3F*, *Figure 3—figure supplement 1B*). Altogether, these results indicate that depletion of TRF1 increases the binding of SUZ12 and the deposition of H3K27me3 at the TSS of genes that are downregulated when TRF1 is depleted, with a particular enrichment in pluripotency- and differentiation-related genes, thus reinforcing the notion that one of the mechanisms by which TRF1 influences pluripotency is through the polycomb repressive complex.

Similarly, we created a Venn diagram to overlap genes bound by SUZ12 and H3K27me3 specifically in the absence of TRF1 with the set of genes that are upregulated in the RNA-seq in this same condition (*Figure 4A*). Interestingly, we found a set of upregulated genes that were only bound by SUZ12 and H3K27me3 in the absence of TRF1. An example of such recruitment in gene *Cxcl12* is shown (*Figure 4B*). We confirmed the increased expression of some of these genes by q-PCR in three independent experiments of TRF1 abrogation (*Figure 4C*). As for the downregulated genes, to obtain more accurate information on SUZ12 and H3K27me3 binding to the genes that are upregulated in the absence of TRF1, we created heat maps of the reads distribution of SUZ12 and H3K27me3 within 2.5 Kb of the transcription start sites (TSS) of all of the genes that were significantly upregulated in the RNA-seq upon TRF1 abrogation (*Figure 4D*). Surprisingly, we observed recruitment of SUZ12 to a great number of upregulated genes (Clusters 1 and 2 of the SUZ12 heatmap (*Figure 4D*)). By contrast, recruitment of H3K27me3 was not so dramatic. We performed Enrichr analysis of the genes contained in Cluster 1 of the SUZ12 heatmap, which corresponds to the larger increase in SUZ12 recruitment. The results showed that upregulated genes contained in Cluster 1 are direct targets of differentiation factors, as well as pluripotency factors and components of polycomb complexes (*Figure 4E*). Importantly, these upregulated genes are enriched in pathways involved in cell differentiation and pluripotency (*Figure 4—figure supplement 1A*). Similar results were obtained when we analyzed genes in which H3K27me3 is recruited to the TSS (*Figure 4F*, *Figure 4—figure supplement 1B*). These results indicate that depletion of TRF1 induces the recruitment of SUZ12 and the deposition of H3K27me3 to genes that are upregulated and that are involved in pluripotency and differentiation. These results are in agreement with previous work showing that SUZ12 is required for mouse embryonic stem cell differentiation and that

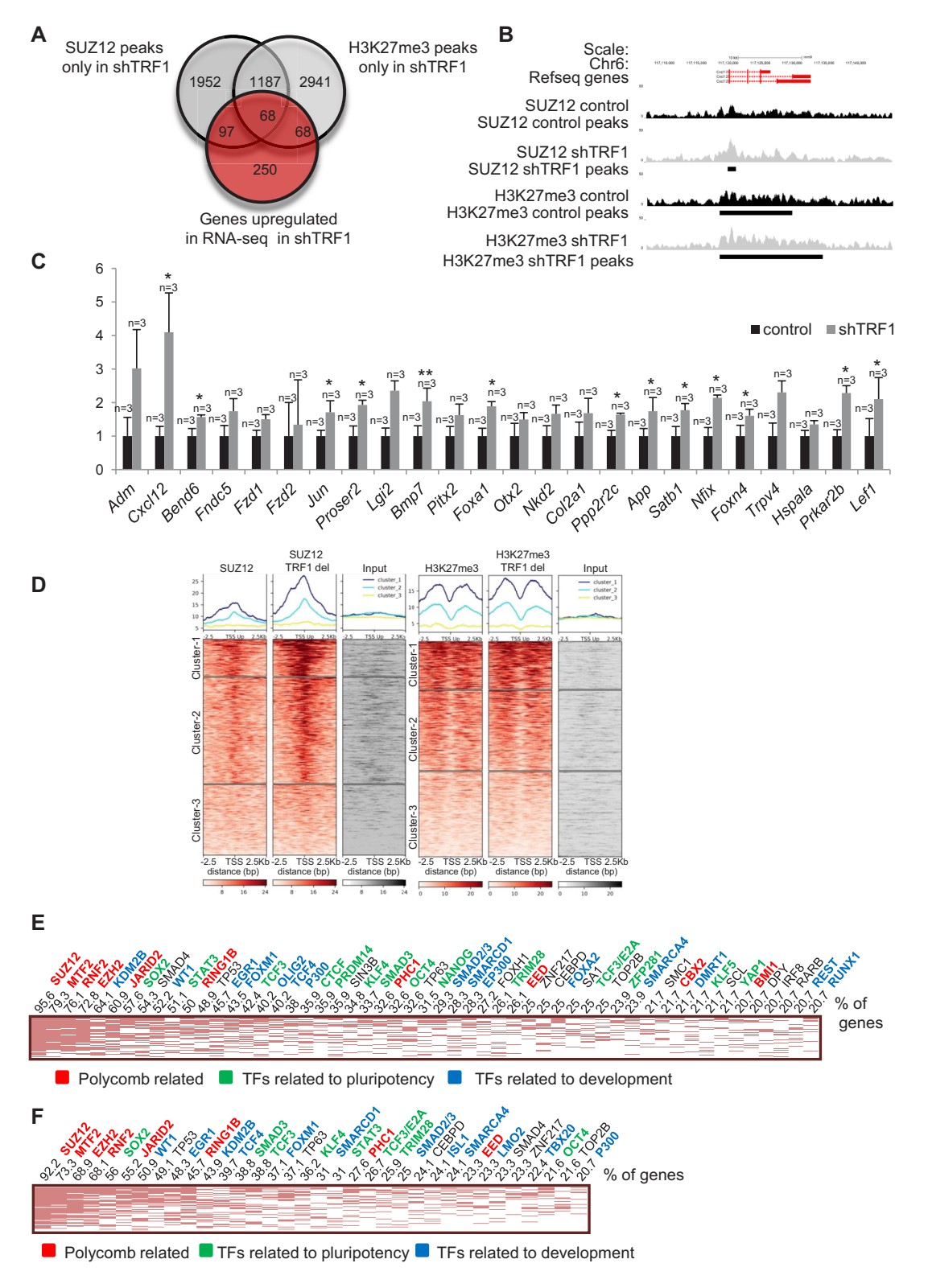

**Figure 4.** Depletion of TRF1 induces the recruitment of SUZ12 and H3K27me3 to genes that are upregulated in the absence of TRF1. (**A**) Venn diagram showing the overlap of genes bound by SUZ12 and H3K27me3 specifically in the absence of TRF1 with the set of genes that are upregulated in this same condition, as obtained by RNA-seq. Note that a number of upregulated genes gain SUZ12 and H3K27me3 peaks upon TRF1 abrogation. (**B**) Gain of SUZ12 and H3K27me3 peaks at the *Cxcl12* gene. (**C**) Gene expression analysis by q-PCR of genes that are upregulated upon TRF1 depletion in which
*Figure 4 continued on next page*

*Figure 4 continued*

SUZ12 or H3K27me3 recruitment was detected. Note that abrogation of TRF1 induces an increase in the expression of these genes, as observed previously by RNA-seq. n = number of independent experiments. Error bars = SE. Statistical analysis was carried out by one -tailed, paired Student's t-test. * = p value<0.05; ** = p value<0.01. (D) Heat maps of the reads distribution of SUZ12 and H3K27me3 within 2.5 Kb of the transcription start sites (TSS) of all of the genes that are significantly upregulated in the RNA-seq upon TRF1 abrogation. Note that a great number of upregulated genes (Clusters 1 and 2 of the SUZ12 heat map) show a clear increase in SUZ12 binding upon TRF1 abrogation, especially in Cluster 1 where the increase is dramatic. By contrast, only Cluster 1 of the H3K27me3 heatmap shows a moderate recruitment. Genes belonging to Cluster 1 of the SUZ12 heatmap (*Figure 4D*), which present a dramatic enrichment of SUZ12 upon TRF1 depletion, were analyzed. (E) The clustergram shows that these genes are targets of pluripotency factors (green), differentiation factors (blue) and encode components of the polycomb (red). Genes belonging to Cluster 1 of the H3K27me3 heatmap (*Figure 4D*), which present a moderate enrichment of H3K27me3 upon TRF1 depletion, were analyzed. (F) As in the case of SUZ12, the clustergram shows that an important number of these genes are targets of pluripotency factors (green), differentiation factors (blue) and components of the encode Polycomb (red).
DOI: https://doi.org/10.7554/eLife.44656.013

The following figure supplement is available for figure 4:

**Figure supplement 1.** Functional analysis of genes that are upregulated in the absence of TRF1 and that show recruitment of SUZ12 and H3K27me3.
DOI: https://doi.org/10.7554/eLife.44656.014

increased levels of PRC2 accumulate on a subset of genes during differentiation, despite their transcriptional activation (*Pasini et al., 2007*). It has been reported that SUZ12 binding to the genome of pluripotent cells in a naïve state is at a low level, which increases upon transferring these cells to serum (primed state) (*Marks et al., 2012*). Thus, we wondered whether the increased recruitment of SUZ12 to the genome upon depletion of TRF1 could represent the loss of the naïve state and a transition to a primed or differentiated state. In primed conditions, SUZ12 is bound to a set of so-called bivalent genes, including the *Hox* clusters (*Voigt et al., 2013*). Thus, we created heatmaps of the reads of SUZ12 and H3K27me3 within 2.5 Kb of the TSS of the described list of bivalent genes (*Marks et al., 2012*) (*Figure 2—figure supplement 3*). We found a dramatic increase of SUZ12 deposition at the TSS of bivalent genes, and a more moderate increase in H3K27me3 at these sites (*Figure 2—figure supplement 3A*). *Figure 2—figure supplement 3B* shows an example of this recruitment at some of the more relevant bivalent genes, the *Hox* clusters. We found that SUZ12 is absent from the *Hox* clusters when TRF1 is present, but is clearly recruited to them in the absence of TRF1. This finding is consistent with the idea that abrogation of TRF1 induces the loss of the naïve state. Also, analysis of the genes from Cluster 1 of the H3K27me3 heatmap in *Figure 4D* (genes that show a moderate increase in H3K27me3) shows that almost 70% of them belong to the group of bivalent genes. It has been reported that bivalent genes show a lower presence of H3K27me3 in naïve-state cells than in primed-state cells (*Marks et al., 2012*), once again supporting the idea that the depletion of TRF1 induces a transition from naïve to primed-differentiated state.

To further understand whether the contribution of TRF1 to polycomb localization was direct or indirect, we performed a chromatin immunoprecipitation of TRF1 followed by deep sequencing (ChIP-seq) in control 2i-grown iPS cells. We observed few peaks of TRF1 when comparing two replicates of the ChIP-seq experiment (*Figure 2—figure supplement 4A*). Comparison of the RNA-seq experiments with TRF1 ChIP-seq indicated that only three genes that are bound by TRF1 showed significantly altered expression when TRF1 was downregulated (*Figure 2—figure supplement 4A* and *Figure 2—figure supplement 4C*). These genes were *Lama1*, which was significantly downregulated when TRF1 was deleted in our RNA sequencing experiment (*Figure 3C*), and *Pitx2* and *Fam43a*, which were significantly upregulated when TRF1 was depleted (*Figure 4C* and Table S1). Interestingly, the majority of the TRF1 peaks outside the telomere were located at extra-telomeric repetitions (TTAGGG or CCCTAA) (*Figure 2—figure supplement 4B*). Enrichr analysis of the genes that are associated with TRF1 peaks revealed that four of these genes (*Figure 2—figure supplement 4B*, labeled in bold), were targets of ZFP322A, a protein that is expressed in the *Inner Cell Mass*, which is essential for ES cell pluripotency and that binds to *Oct4* and *Nanog* promoters and regulates their transcription (*Ma et al., 2014*). These four genes were present among the 100 binding sites with top-ranked peak heights in ZFP322A ChIP-seq analysis (*Ma et al., 2014*). Most of the TRF1 peaks are located in intergenic regions, but in all four of the genes targetted by ZFP322A, the TRF1 peaks were located in the introns. Two of these genes (*Pde1c* and *Ppp1r9a*) were upregulated in the RNA-seq and one of them, *Lama1*, was significantly downregulated (*Figure 2—figure*

*supplement 4C*), whereas expression of the fourth gene (*Bbox1*) was not detected. Interestingly, ZFP322A controls the expression of *Sall4* and *Zscan10*, two genes that are downregulated upon TRF1 deletion (see *Figure 3C*). These results suggest that TRF1 may directly regulate ZFP322A, which in turn, regulates key genes involved in pluripotency, although the mRNA levels of this gene were not altered by TRF1 abrogation in the ChIP-seq data (not shown). However, although TRF1 is able to bind a few locations in the genome, this binding does not seem sufficient to explain the vast epigenetic changes that we observed in the absence of TRF1.

In order to determine whether TRF1 directly regulates the binding of polycomb in the genome, we performed mass spectroscopy of protein complexes that were immunoprecipitated with an anti-GFP antibody in *Terf1^GFP/GFP^*, *Terf1^+/GFP^* and *Terf1^+/+^* iPS cells (*Schneider et al., 2013*). We identified 64 proteins that showed co-immunoprecipitation in both *Terf1^GFP/GFP^* and *Terf1^+/GFP^* cells but not in *Terf1^+/+^* cells. We confirmed the presence in this group of proteins of different components of shelterin, namely TPP1, RAP1, TRF1, TIN2 and POT1. We did not, however, detect the presence of SUZ12 in the immunocomplex, suggesting that there is no direct interaction between TRF1 and PRC2, at least in this experimental setting (*Supplementary file 4*).

## Depletion of TRF1 in 2i-grown iPS cells induces the upregulation of TERRA RNAs expression

Interestingly, the telomeric TERRA RNAs have been recently shown to bind throughout the genome in H3K27me3-rich regions, and to interact with components of the polycomb complex (*Chu et al., 2017*). In our group, we also recently showed that TERRA proteins are required to recruit polycomb to telomeres and so contribute to H3K27me3 deposition at telomeric chromatin (*Montero et al., 2018*). With these findings in mind, we set to address whether the expression of TERRA was altered when TRF1 was depleted, which could explain, at least in part, the effects of TRF1 depletion on polycomb. First, we performed RNA-FISH analysis using a TERRA-specific probe in control 2i-grown iPS cells and in 2i-grown iPS cells in which TRF1 expression had been reduced. The results showed a dramatic upregulation of TERRA signal in TRF1-depleted cells (*Figure 5A*). Northern-blot analysis of TERRA levels in three independent TRF1 abrogation experiments in iPS cells showed TERRA upregulation as a consequence of TRF1 downregulation (*Figure 5B*). Note that TERRA is normally increased in iPS cells compared to differentiated MEFs cells, as we have previously described (*Marion et al., 2009*), and that TERRA levels are further increased by TRF1 abrogation in iPS cells (*Figure 5C*), clearly indicating that TRF1 has a role as a repressor of TERRA during the induction of pluripotency. Finally, we also confirmed increased TERRA expression as the result of TRF1 abrogation in keratinocytes from *Terf1^Δ/Δ^ K5-Cre* newborn mice, which lack TRF1 expression (*Figure 5D*). These results are in agreement with previous findings from other groups in human cells (*Porro et al., 2014*; *Zeng et al., 2017*; *Sadhukhan et al., 2018*), which also show TERRA increase upon TRF1 depletion. Other studies, however, showed that depletion of TRF1 in mouse immortalized MEFs did not increase TERRA levels (*Sfeir et al., 2009*), and that downregulation of TRF1 by siRNA in the immortalized mouse myoblast cell line C2C12 reduced the expression of TERRA (*Schoeftner and Blasco, 2008*). A possible explanation for this apparent contradiction could be that the immortalization process may alter TERRA expression or that TERRA regulation by TRF1 is cell-type specific. In any case, the results shown here clearly demonstrate a dramatic increase in TERRA in two different mouse cells types (iPS cells and keratynocytes) upon TRF1 depletion (*Figure 5*), which in turn could contribute to the recruitment of polycomb to new parts of the genome.

Previous studies have demonstrated that the depletion of the shelterin component TRF2 can also induce higher TERRA expression (*Porro et al., 2014*; *Rossiello et al., 2017*). We wondered whether the global changes in gene expression observed upon TRF1 abrogation in 2i-grown iPS cells were also detected when depleting TRF2 or whether they were specific for TRF1 depletion. To this end, we downregulated the expression of TRF2 by means of an shRNA (*Figure 5—figure supplement 1A*) in *Trp53^-/-^*2i-grown iPS cells and measured TERRA levels by RNA-FISH analysis, using a TERRA-specific probe (*Figure 5—figure supplement 1B*), and by Northern Blot (*Figure 5—figure supplement 1C*). However, in both cases we found that depletion of TRF2 in our experimental system did not induce an increase in TERRA levels (*Figure 5—figure supplement 1B–C*), supporting the idea that, at least in 2i-grown iPS cells, TRF2 does not have a major role in controlling TERRA expression.

We have previously shown that TERRA is upregulated during the induction of iPS cells (*Marion et al., 2009*) (see also *Figure 5*). We wondered whether TERRA levels were also regulated

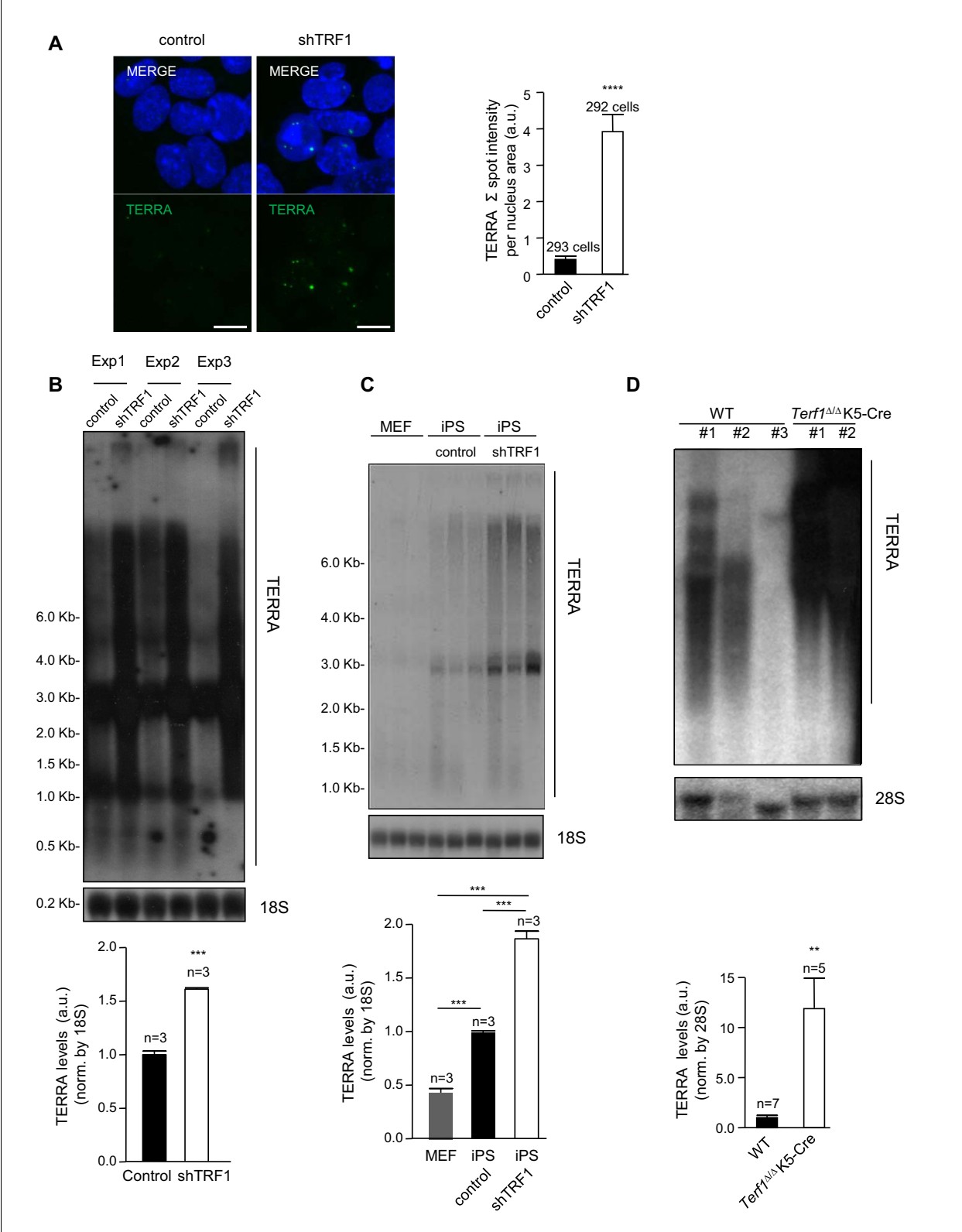

**Figure 5.** Abrogation of TRF1 induces the upregulation of TERRA RNAs expression. (**A**) Left, representative image of RNA-FISH using a TERRA-specific probe in control 2i-grown iPS and in 2i-grown iPS cells in which TRF1 is depleted. Note the dramatic increase in TERRA signal in cells lacking TRF1. Right, quantification of the RNA-FISH images. Data were obtained from one experiment. n = number of cells analyzed from each sample. Error bars = SE. Statistical analysis was carried out by Student's t-test. **** = p-value <0.0001. (**B**) Top, representative image of a Northern-blot analysis of

*Figure 5 continued on next page*

*Figure 5 continued*

TERRA RNAs expression in control 2i-grown iPS and in 2i-grown iPS cells in which TRF1 is depleted in three independent experiments. Note the clear upregulation of TERRA levels in cells lacking TRF1, confirming the RNA-FISH data. Bottom, quantification of TERRA levels from the Northern blot, normalized to 18S levels. Error bars = SE. Statistical analysis was carried out by Student's t-test. *** = p-value <0.0001. (C) Top, image of a Northern-blot analysis of TERRA RNAs expression in MEF, 2i-grown iPS and 2i-grown iPS depleted for TRF1. Bottom, quantification of TERRA levels from the Northern blot, normalized to 18S levels. n = number of independent experiments. Error bars = SE. Statistical analysis was carried out by Student's t-test. (D) Top, representative image of a Northern-blot analysis of TERRA RNAs expression in keratinocytes of wild-type or *Terf1*$^{\Delta/\Delta}$ *K5-Cre* (lacking expression of TRF1) newborn mice. Again, note the clear upregulation of TERRA levels in cells lacking TRF1. Bottom, quantification of TERRA levels from the Northern blot, normalized to 28S levels. n = number of independent newborn mice analyzed. Error bars = SE. Statistical analysis was carried out by Student's t-test.

DOI: https://doi.org/10.7554/eLife.44656.015

The following figure supplements are available for figure 5:

**Figure supplement 1.** TRF2 abrogation in *Trp53*$^{-/-}$ 2i-grown iPS cells does not change TERRA levels.

DOI: https://doi.org/10.7554/eLife.44656.016

**Figure supplement 2.** TRF1 protein and TERRA levels decrease upon differentiation of ES cells.

DOI: https://doi.org/10.7554/eLife.44656.017

during the process of differentiation. To answer this question, we differentiated ES cells with retinoic acid and measured TERRA levels by RNA-FISH after 5 days of treatment (*Figure 5—figure supplement 2A*). In agreement with loss of pluripotency, we confirmed significantly decreased *Nanog* levels after 5 days of treatment with retinoic acid (*Figure 5—figure supplement 2A*). We found that TERRA expression decreased during cellular differentiation (*Figure 5—figure supplement 2B*). We also found that upon differentiation of ES cells with retinoic acid, the levels of TRF1 protein were decreased, as measured by immunofluorescence (*Figure 5—figure supplement 2C*), in accordance with previous data showing higher levels of TRF1 in pluripotent cells (*Marion et al., 2009*; *Schneider et al., 2013*). These findings suggest a fine-tuned regulation of TERRA levels during development, which is most probably linked to cell identity.

## Genomic TERRA binding correlates with genes that are differentially expressed in the absence of TRF1 and at which SUZ12 and H3K27me3 are present

In order to address a role of TERRA in PRC2 localization to genes, we analyzed TERRA binding throughout the whole genome in our experimental setting. To this end, we performed TERRA CHIRT sequencing (CHIRT-seq) (as described by *Chu et al., 2017*), in 2i-grown *Trp53*-null iPS cells. We found that TERRA was able to bind to 10,670 locations across the genome with an enrichment fold equal to or higher than 10 when compared to the input. Using Homer software, TERRA CHIRT peaks with an enrichment fold of 10 or higher were assigned to genes and genome regions. We found that the vast majority of TERRA binding sites were localized in non-coding regions (mostly in intergenic or intronic regions) (*Figure 6A*), in agreement with previous findings in embryonic stem cells (ESC) (*Chu et al., 2017*). Importantly, we found that 90.4% of the genes previously associated to TERRA peaks in ESC (*Chu et al., 2017*) were also found in our CHIRT-seq with 2i-grown iPS cells (*Figure 6B*), thus validating both TERRA binding sites along the genome and our CHIRT experiment. Importantly, we found that the TERRA peaks with a higher fold enrichment were localized at the subtelomere of chromosome 18 (*Figure 6C*), a locus previously described by us as one of the *bona fide* TERRA loci in murine cells (*López de Silanes et al., 2014*), thus further validating the importance of this locus in TERRA transcription. Also, the peaks with a higher fold enrichment colocalized with the subtelomeric sequence of TERRA at chromosome 18 (*Figure 6C*), confirming that we can capture the cis interaction of TERRA with the chromatin by using CHIRT. Altogether, these data clearly show that the CHIRT-seq technique was able to map TERRA localizations in the genome, including those at chromosome 18 subtelomere that were previously described by us as one of the main origins of mouse TERRA.

We next analyzed the genes that are associated with TERRA peaks using Enrichr. We found that more than 40% of these genes were targets of SUZ12, and that they were also bound by several components of the PRC2 complex (*Figure 6D*), supporting the idea that TERRA and PRC2 bound to similar locations in the genome. Furthermore, most of those genes were bound by transcription

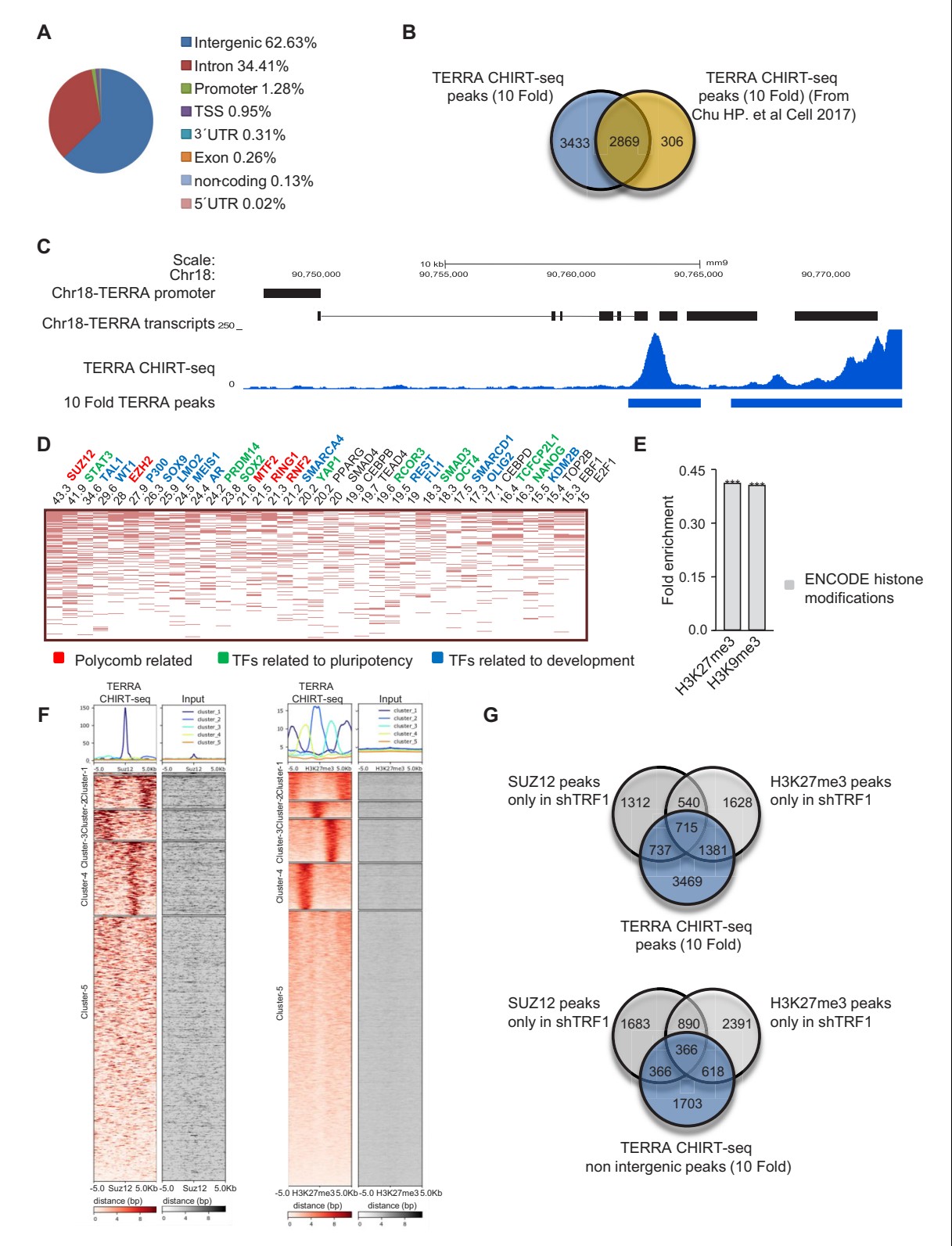

**Figure 6.** TERRA CHIRT-seq analysis. (**A**) Genome-wide TERRA localization in different genome regions. (**B**) Comparison of the genes associated with TERRA peaks in this work and in previously described work (*Chu et al., 2017*). Note that the majority of the genes found in the previous study are present in our CHIRT-seq. (**C**) Representation of the peak with higher enrichment in TERRA CHIRT-seq. Note that this peak coincides with a TERRA locus previously described at chromosome 18 (*López de Silanes et al., 2014*). (**D**) CHEA analysis of genes annotated to TERRA peaks. (**E**) Encode

*Figure 6 continued on next page*

*Figure 6 continued*

histone modifications analysis of genes annotated to TERRA peaks. (F) Heatmaps of TERRA reads within 5 Kb of SUZ12 and H3K27me3 peaks. (G) Top, overlapping of genes annotated to TERRA peaks with the peaks of SUZ12 and H3K27me3 exclusive for the shTRF1 ChIP-seq sample. Bottom, overlapping of genes annotated to TERRA peaks in non intergenic regions, with peaks of SUZ12 and H3K27me3 exclusive for the shTRF1 ChIP-seq sample. The experiment was performed once.

DOI: https://doi.org/10.7554/eLife.44656.018

factors that are related to pluripotency or differentiation (*Figure 6D*), suggesting that TERRA could be important for the loss of pluripotency after TRF1 depletion, through PRC2 regulation. In agreement with this, those genes were enriched in the H3K27me3 mark (*Figure 6E*).

We then performed heatmaps of the TERRA CHIRT-seq reads within 5 Kb of the peaks from SUZ12 and H3K27me3 ChIP-seq (*Figure 6F*). We found that 34.4% of the SUZ12 peaks and 32.6% of the H3K27me3 peaks have TERRA binding in their vicinity (*Figure 6F*). To understand whether the gain of SUZ12 and H3K27me3 binding after TRF1 removal is associated to locations where TERRA is present, we compared the genes that are associated with TERRA peaks with the genes associated with SUZ12 and H3K27me3 peaks that are exclusive for the shTRF1 ChIP-seq (*Figure 6G*). We found that 43.9% of the genes annotated to SUZ12 peaks were exclusive for shTRF1 ChIP-seq and that 49.2% of those from H3K27me3 peaks were also annotated in the TERRA CHIRT-seq. The fact that only a fraction of the newly recruited sites for SUZ12 coincides with TERRA peaks could be explained if SUZ12 recruitment to the genome were to be induced in part by the secondary changes in global epigenetic status and gene expression generated by TRF1 depletion. Also, the chromatin sonication process required for the ChIP and CHIRT techniques may be disrupting the long-range interaction mediated by TERRA. All together, these data indicate that there is a correlation of TERRA location and SUZ12 and H3K27me3 that includes the genes in which SUZ12 and H3K27me3 are deposited after TRF1 depletion, suggesting that TERRA could be mediating the effects of TRF1 abrogation on polycomb-regulated genes.

To evaluate whether the presence of TERRA binding could be associated with the changes of gene expression observed after TRF1 depletion, we compared the genes that were significantly altered in our RNA-seq with the genes that are associated with TERRA peaks (*Figure 7A*). We found that 29.9% of the genes that are downregulated and 33.5% of the genes that are upregulated when TRF1 was deleted are annotated to TERRA peaks. Then, we created heatmaps of the TERRA CHIRT-seq reads within 2.5 Kb of the TSS of the downregulated genes (*Figure 7B*) and the upregulated genes (*Figure 7C*). We observed that 29.1% of the downregulated genes (Clusters 1 and 2 *Figure 7B*) and 27% of the upregulated genes (Clusters 1 and 2 *Figure 7C*) are bound by TERRA in the proximity of the TSS. To determine whether those genes have increased binding of SUZ12 or an increased number of H3K27me3 marks after TRF1 removal, we compared the downregulated and upregulated genes that present TERRA signal (*Figure 7B–C*) with the genes with increased SUZ12 or H3K27me3 signal (from *Figure 3D* and *Figure 4D*). The results confirmed that 90.1% of the downregulated genes bound by TERRA and 70.1% of the upregulated ones have increased SUZ12 or H3K27me3 marks (as determined by the reads of the heatmaps) (*Figure 7D,E*). As shown in the UCSC genome browser representation of the SUZ12 and H3K27me3 CHIP-seq and the TERRA CHIRT-seq of the downregulated gene *Pou2f3* and of the upregulated gene *Bmp7* (*Figure 7D,E*), the TERRA peaks are present at the promoters of the genes and close to the increased SUZ12 or H3K27me3 signal. These data suggest that a significant number of the transcriptional changes observed after TRF1 deletion could be due to the recruitment of PRC2 complex by TERRA.

Next, we analyzed the genes that are co-regulated by the PRC2 complex and TERRA using Enrichr. We found that genes that were downregulated when TRF1 was depleted and that had increased binding of PRC2 and TERRA (*Figure 7D*) are mostly targets of important transcription factors involved in pluripotency, such as OCT4, SOX2 or NANOG (*Figure 7F*). Also, they are significantly enriched in the PluriNetwork pathway from the Wikipathways database (*Figure 7G*). In the case of genes that are upregulated when TRF1 was depleted and that had increased binding of PRC2 and TERRA (*Figure 7E*), the Enrichr analysis revealed that they were targets of several differentiation transcription factors, SUZ12 and other PRC2 complex components, and to a less extent, of transcription factors involved in pluripotency (*Figure 7H*). Moreover, those genes are enriched in the Neural Crest Differentiation pathway from Wikipathways (*Figure 7I*) and in genes in which the

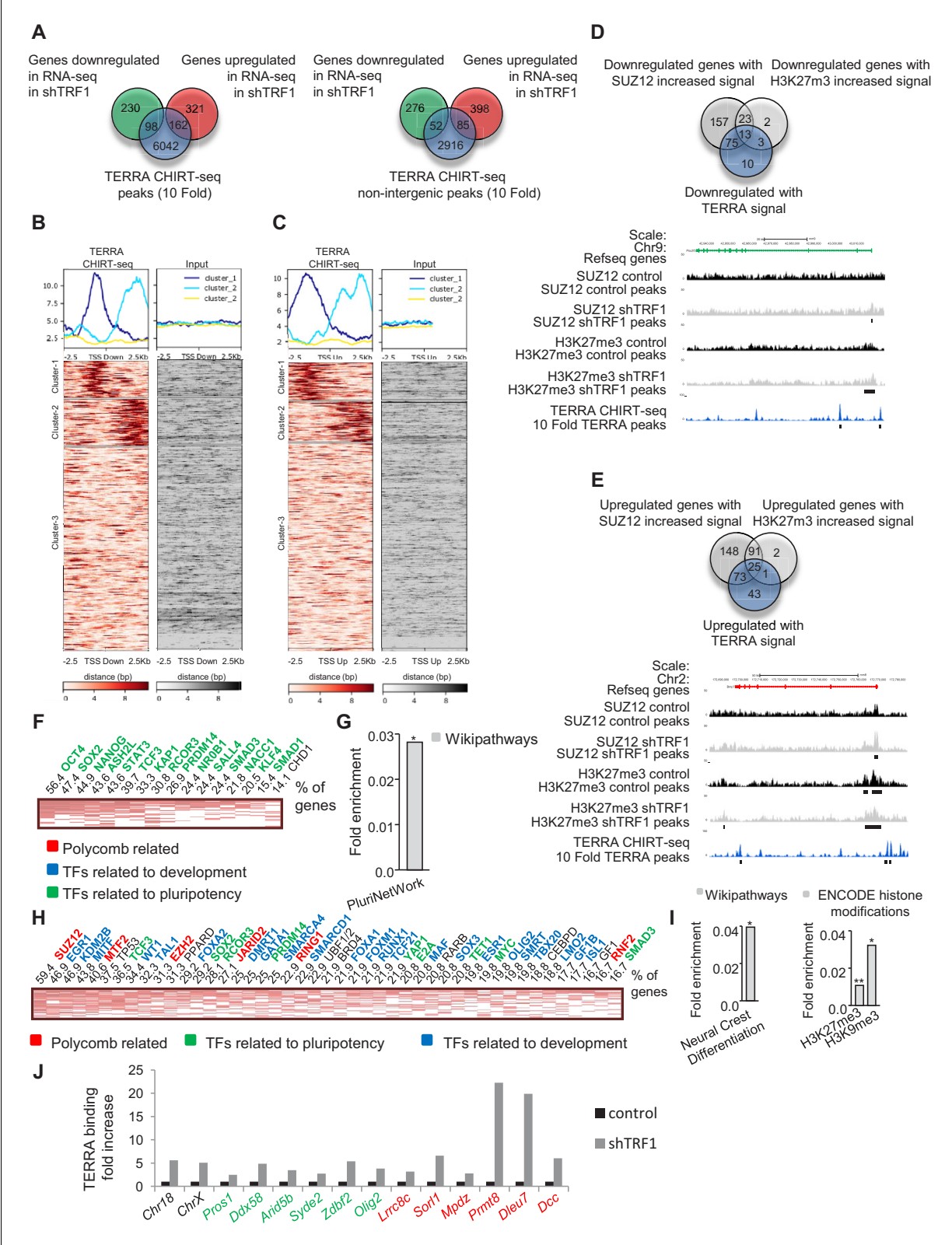

**Figure 7.** TERRA binding correlates with genes that are differentially expressed in the absence of TRF1 and with the presence of SUZ12 and H3K27me3 and is increased upon TRF1 depletion. (A) Left, Venn diagram showing the overlap between the genes annotated to TERRA peaks and genes that are up- or downregulated in the RNA-seq after TRF1 depletion. Right, Venn diagram showing the overlap between the genes annotated to TERRA peaks in non-intergenic regions and genes that are up- or downregulated in the RNA-seq after TRF1 depletion. (B) Heatmaps showing TERRA CHIRT-

*Figure 7 continued on next page*

*Figure 7 continued*

seq reads within 2.5 Kb of TSS of genes downregulated after TRF1 depletion. (C) Heatmaps showing TERRA CHIRT-seq reads within 2.5 Kb of TSS of genes upregulated after TRF1 depletion. (D) (Top) Overlapping between downregulated genes with TERRA signal and downregulated genes in which the SUZ12 or H3K27me3 signal is increased. (Bottom) Representative image of SUZ12, H3K27me3 and TERRA reads and peaks in a downregulated gene. (E) (Top) Overlapping between upregulated genes with TERRA signal and upregulated genes in which the SUZ12 or H3K27me3 signal is increased. (Bottom) Representative image of SUZ12, H3K27me3 and TERRA reads and peaks in an upregulated gene. (F) CHEA analysis of genes downregulated in the RNA-seq after the deletion of TRF1 that are bound by TERRA and that show increased SUZ12 and/or H3K27me3 signal. (G) Wikipathways analysis of genes from panel (F). (H) CHEA analysis of genes upregulated in the RNA-seq after deletion of TRF1 that are bound by TERRA and that show increased SUZ12 and/or H3K27me3 signal. (I) Wikipathways analysis and Encode histone modification analysis of genes from panel (H). (J) CHIRT-qPCR in control and shTRF1 samples from one experiment in genomic regions previously identified as TERRA binding sites in the CHIRT-seq. Genomic regions labeled in black correspond to subtelomeric regions where TERRA is potentially transcribed. Genomic regions labeled in green or red correspond to regions close to the promoters of genes that are downregulated or upregulated, respectively, when downregulating TRF1. Note that, in all the cases, TERRA binding is clearly increased upon TRF1 abrogation. Values were normalized to inputs and to a genomic region where TERRA binding was not found in the CHIRT-seq.

DOI: https://doi.org/10.7554/eLife.44656.019

ENCODE histone modifications database shows the presence of H3K27me3 and H3K9m3 marks (*Figure 7I*). These data indicate that the possible recruitment of PRC2 by TERRA after TRF1 depletion could alter a set of genes that can initiate the process of naïve state loss and differentiation.

## TRF1 depletion induces increased TERRA binding to genes with increased polycomb binding and altered gene expression upon TRF1 abrogation

To test whether TRF1 abrogation resulted in increased recruitment of PRC2 by TERRA, we selected a number of genomic regions close to gene promoters where we had previously found both binding of TERRA in our CHIRT-seq and increased binding of PRC2 and changes in gene expression upon TRF1 depletion. We performed CHIRT-qPCR analysis for these regions in both 2i-grown iPS depleted for TRF1 and in non-depleted controls (*Figure 7J*). As a control, we included two genomic regions where TERRA is potentially transcribed (*Figure 7J*, labeled in black). We found a clear increase in TERRA binding to all of these regions upon TRF1 depletion, including the control regions where TERRA is potentially transcribed (suggesting that the increased levels of TERRA upon TRF1 deletion result from higher levels of TERRA transcription), the downregulated genes (*Figure 7J*, labeled in green) and the upregulated genes (*Figure 7J*, labeled in red). These results indicate both higher TERRA binding to genes with increased binding of PRC2 and changes in gene expression upon TRF1 depletion, suggesting that the higher TERRA binding could mediate the increased SUZ12 recruitment in these regions, and could thus control the expression of genes that are important for pluripotency and differentiation (see Model in (*Figure 8*).

As higher levels of TERRA mediate the increased recruitment of SUZ12 to the genome, we wondered whether TRF1 depletion would increase the interaction between TERRA and SUZ12. We therefore performed an RNA immunoprecipitation (RIP) experiment, in which we immunoprecipitated SUZ12 in both control and TRF1-depleted *Trp53*$^{-/-}$ 2i-grown iPS cells, and measured the amount of TERRA that was pulled-down with SUZ12. The results indicate that, indeed, SUZ12 shows a higher level of binding to TERRA when TRF1 is depleted (*Figure 9*), reinforcing the idea that higher levels of TERRA mediate the recruitment of SUZ12 to the genome.

## Discussion

In recent years, mounting evidence has suggested a role of the TRF1 telomere binding protein in the acquisition and maintenance of pluripotency and stemness, which seemed to be additional to its known role in maintaining telomere protection and preventing telomere fusions. In particular, we and others found that TRF1 is highly upregulated in ES cells and iPS cells (*Boué et al., 2010*; *Schneider et al., 2013*). Our group further showed that *Terf1* is a direct target of the pluripotency gene *OCT4* and that the upregulation of *Terf1* is an early event during the induction of pluripotency (*Schneider et al., 2013*). Indeed, we found that TRF1 is needed for both the induction and the maintenance of pluripotency, being essential therefore for nuclear reprogramming

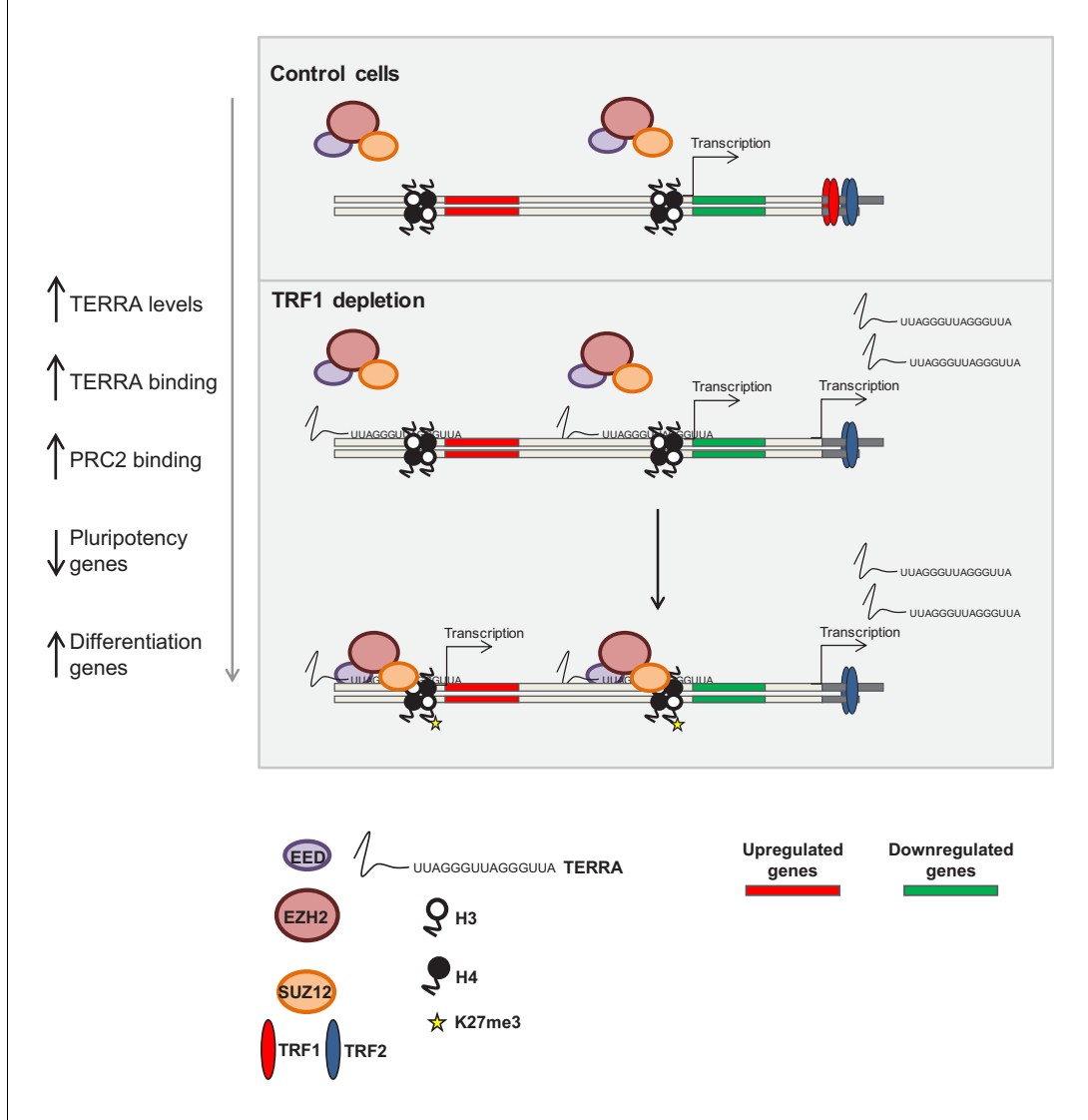

**Figure 8.** Model of the role of TRF1 in controlling pluripotency through TERRA expression and TERRA-dependent polycomb recruitment. In control 2i-grown iPS cells (top) the levels of TRF1 are elevated, the PRC2 complex is barely bound to the genome and pluripotency genes are expressed. When TRF1 levels are downregulated (bottom), TERRA expression is greatly increased, resulting in higher levels of binding to the genome. In this way, TERRA could increase PRC2 recruitment to many locations within the genome that are involved in the control of pluripotency and differentiation.
DOI: https://doi.org/10.7554/eLife.44656.020

(*Schneider et al., 2013*). In addition, we demonstrated that during in vivo reprogramming, TRF1 expression is highly upregulated in the de-differentiated areas upon reprogramming and that TRF1 chemical inhibition reduced the efficiency of in vivo reprogramming (*Marión et al., 2017*). TRF1 is also present at high levels in the Inner Cell Mass (ICM) of the blastocyst when compared to differentiated mouse embryonic fibroblasts (MEFs) (*Varela et al., 2011*). Together these findings suggested a role for TRF1 upregulation during in vitro and in vivo tissue reprogramming and pluripotency (*Marión et al., 2017*). However, the molecular mechanisms through which TRF1 modulates pluripotency has not yet been revealed.

Here, we make the unprecedented finding that TRF1 controls the maintenance of the pluripotency state by influencing the epigenetic status of the chromatin. In particular, TRF1 depletion in naïve-state pluripotent stem cells leads to dramatic changes in the expression of genes that are associated with pluripotency pathways, as well as of genes that are controlled by the polycomb complex PRC. These changes are consistent with loss of the naïve state as the PRC2 complex

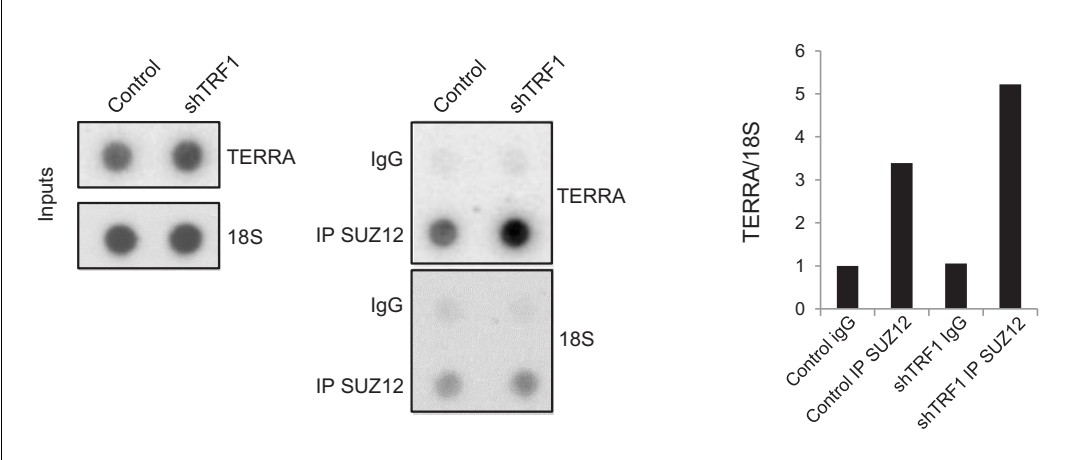

**Figure 9.** SUZ12 binding to TERRA increases upon TRF1 depletion. Immunoprecipitation of SUZ12 and detection of bound TERRA by dot-blot hybridization. Left, representative images of the input and immunoprecipitated TERRA in control cells and in cells depleted for TRF1. Note that levels of TERRA are higher in the SUZ12 immunocomplex when TRF1 is abrogated. Right, quantification of the TERRA and 18S levels in the immunocomplex. Levels of TERRA were normalized by the levels of 18S. Note again that binding of SUZ12 increases upon TRF1 depletion. The experiment was performed once.

DOI: https://doi.org/10.7554/eLife.44656.021

is recruited to bivalent genes and the differentiation programs are initiated. We further demonstrate that TRF1 depletion leads to increased binding of PRC2 complex to its target genes, which is concomitant with an increase in the abundance of the H3K27me3 polycomb mark at some of these sites. Accordingly, we see that TRF1 depletion induces changes in the expression of the genes in which these TRF1-dependent PRC2 sites are located.

In *Arabidopsis thaliana*, telomere-repeat binding factors (TRBs) similar to TRF1 have been reported to recruit PRC proteins to different promoters through a telobox motif (*Zhou et al., 2016b*; *Zhou et al., 2018*). Although the binding of human TRF1 to extratelomeric sites had been reported (*Simonet et al., 2011*), we show here for the first time that mouse TRF1 can directly bind to a set of genes containing TTAGGG/AATCCC repeats in iPS cells in a naïve state. Interestingly, a significant percentage of these genes are targets of the pluripotency regulator ZFP322A, opening the possibility that TRF1 may also exert a direct effect on pluripotency by directly regulating the transcription of these genes. However, the vast epigenetic changes observed here as the consequence of TRF1 depletion, which involve the re-localization of PRC2 to many genomic sites, are not likely to be explained by the binding of TRF1 to these few extra-telomeric sites.

TRF1 has been previously shown to interact with the telomere-originated long non-coding RNAs known as TERRA (*Deng et al., 2009*; *Chu et al., 2017*). We have previously shown that TERRA interacts with polycomb components and that this interaction is important to establish polycomb marks and heterochromatin marks at telomeric chromatin (*Montero et al., 2018*). Here, we demonstrate that TERRA can also bind to PRC2 sites in the genome that are dependent on TRF1. More importantly, we show that TRF1 depletion causes a dramatic increase in TERRA expression, in agreement with previous observations in human cells that show increased levels of TERRA upon TRF1 downregulation (*Porro et al., 2014*; *Zeng et al., 2017*; *Sadhukhan et al., 2018*). This upregulation of TERRA expression coincides with an increased binding of TERRA to genomic regions where PRC2 is also increased upon TRF1 depletion and that are important in controlling pluripotency and differentiation. Interestingly, these TERRA- and PRC2-bound regions also show altered expression upon TRF1 abrogation. Finally, we demonstrate that SUZ12 binding to TERRA increases upon TRF1 depletion, reinforcing the idea that TERRA mediates the recruitment of SUZ12 to the genome.

All together, these findings suggest a model by which TRF1-dependent TERRA upregulation allows TERRA-mediated recruitment of PRC2 to key pluripotency and differentiation genes, thus controlling the epigenetic landscape of pluripotent stem cells (see the model in *Figure 8*). Finally, these findings explain the fact that TRF1 is upregulated during in vitro and in vivo reprogramming

and that it is essential for the reprogramming and maintenance of pluripotency. They also are in agreement with the fact that TERRA is tightly regulated during pluripotency and differentiation.

## Materials and methods

### Cells and culture conditions

The primary keratinocytes and iPS used in this work were generated in our lab and were tested for mycoplasm in a routine manner. $Trp53^{-/-}$, $Terf1^{+/+}$, $Terf1^{+/GFP}$ and $Terf1^{GFP/GFP}$ iPS cells were cultured in gelatin-coated plates with DMEM (high glucose) supplemented with serum replacement (KSR, Invitrogen), LIF 1000 u/ml, non-essential amino acids, glutamax and beta-mercaptoethanol (known as iPS medium). The medium was supplemented with MEK inhibitor PD0325901 (1 μM) and GSK3 inhibitor CHR99021 (3 μM), together known as 2i. Primary keratinocytes from newborn mice were obtained and grown as described in previous work (*Martínez et al., 2009*).

ES cells were plated in complete ES medium (FCS + LIF) on gelatin-coated 35mm plates at a density of $2.5 \times 10^5$ cells per plate. Next day, the medium was changed to ES medium (–LIF) containing 1 μM retinoic acid. Cells were cultured for 24, 48, 72 hr to 5 days changing medium daily.

### TRF1 or TRF2 abrogation in iPS cells

Lentiviral supernatants were produced in HEK-293T cells ($5 \times 10^6$ cells per 100-mm-diameter dish) transfected with the packaging plasmids pMDLg/pRRE (3.25 μg), pRSV.Rev (1.25 μg), pMDG VSVG (1.75 μg) (obtained from Addgene), and one of three shRNA lentiviral constructs (5 μg), pLKO.1-puro-scramble shRNA (obtained from Addgene), pLKO.1-puro-TRF1 shRNA (bacterial glicerol stock (TRCM0000071298, obtained from Sigma-Aldrich) or pLKO.1-puro-TRF2 shRNA (CCTTGGAA TCAGCTATCAATG). Transfections were performed using Fugene-6 transfection reagent (Roche) according to the manufacturer's protocol. Cells were cultured in standard ES cells medium containing LIF. After 2 days, viral supernatants (10 ml) were collected serially during the subsequent 36 hr, at 12 hr intervals, each time adding fresh ES cell medium to the cells (10 ml).

The recipient $Trp53^{-/-}$ iPS cells had been seeded the previous day ($2 \times 10^6$ cells per 100-mm-diameter dish) in 2i-containing iPS medium, and received 4.5 ml of the corresponding viral supernatants (supplemented with 2i) and 5.5 ml of 2i-containing iPS medium. The procedure was repeated three times at 12 hr intervals. 12 hr after the last infection was completed, and the medium was replaced with 2i-containing iPS medium to remove the virus. The following day, the iPS cells were expanded in 2i-containing iPS medium supplemented with puromycin to select for the shRNAs vectors. After 48 hr, samples for the different experiments were collected.

### Telomeric FISH

TRF1 was depleted as described above. Metaphase and Q-FISH hybridization was performed as previously described (*Samper et al., 2001*; *Gonzalo et al., 2006*).

### Western blots

Whole-cell extracts were prepared by resuspending cells ($10 \times 10^6$ cells/ml) in a buffer containing 10 mM Hepes (ph 7.9), 10 mM KCl, 2.5 mM $MgCl_2$, 0.34 M sucrose and 10% glycerol, supplemented with 0.1 mM phenylmethylsulfonyl fluoride (PMSF), protease inhibitors cocktail and Dnase (100 u/ml). After 30 min on ice, extracts were sonicated and resolved on NuPAGE 4–12% gradient Bis-Tris gels. After protein transfer onto nitrocellulose membrane (Whatman), the membranes were incubated with the primary indicated antibodies: rat antibody against TRF1 (raised in our laboratory against full-length mouse TRF1 protein) (1:500), mouse monoclonal against β-actin (A1978 Sigma, 1:5000), rabbit polyclonal antibody against SUZ12 (ab12073, 1:500), rabbit antibody against H3K27me3 (07–449 Upstate, 1:3000), mouse monoclonal antibody against phospho histone H2A.X (Ser139) (Millipore 05–636, 1:200) and rabbit antibody against SMC1 (A300-055A Bethyl, 1:1000). Antibody binding was detected after incubation with a secondary antibody coupled to horseradish peroxidase using chemiluminescence with ECL detection KIT (GE Healthcare). Western blots for SUZ12, H3K27me3 and H2A.X were performed with samples obtained from four independent TRF1 deletion experiments. Statistical analysis was performed by Student's t-test.

## RNA-seq

For RNA-seq analysis, two independent TRF1 deletion experiments were carried out, obtaining two controls samples and two TRF1-deleted samples. Total RNA was extracted with TRIzol reagent (ThermoFisher 15596026) and then purified and treated with DNAse using an RNeasy Kit (QUIAGEN 74106), following the manufacturer's instructions. Total RNA (1 µg) was used for the RNA-seq experiment, and the obtained sample RNA Integrity Number was 10 (Agilent 2100 Bioanalyzer). The poly (A)+ fraction was purified and randomly fragmented, converted to double-stranded cDNA and processed through subsequent enzymatic treatments of end-repair, dA-tailing, and ligation to adapters with the 'NEBNext Ultra II Directional RNA Library Prep Kit for Illumina' (NEB, Cat. No. E7760) as recommended by the manufacturer. This kit incorporates dUTP during second-strand cDNA synthesis, which implies that only the cDNA strand generated during first strand synthesis is eventually sequenced. An adapter-ligated library was completed by PCR with Illumina PE primers. The resulting purified cDNA library was applied to an Illumina flow cell for cluster generation and sequenced on an Illumina NextSeq 500 sequencer, following manufacturer's protocols. Single-end sequenced reads were analyzed with the next*presso* pipeline (*Graña et al., 2018*) as follows. Sequencing quality was checked with FastQC v0.10.1 (http://www.bioinformatics.babraham.ac.uk/projects/fastqc/). Reads were aligned to the mouse reference genome (NCBI37/mm9, https://ccb.jhu.edu/software/tophat/igenomes.shtml) with TopHat-2.0.10 (*Trapnell et al., 2012*) using Bowtie 1.0.0 (*Langmead et al., 2009*) and Samtools 0.1.1.9 (*Li et al., 2009*), allowing three mismatches and 20 multihits. Quantification of transcripts and differential expression were calculated with Cufflinks 2.2.1 (*Trapnell et al., 2012*), using the mouse NCBI37/mm9 transcript annotations from https://ccb.jhu.edu/software/tophat/igenomes.shtml. For differential expression, a false discover rate (FDR) <0.05 was used. GSEAPreranked (*Subramanian et al., 2005*) was used to perform gene set enrichment analysis on a pre-ranked gene list, setting 1000 gene set permutations. Only those gene sets with significant enrichment levels (FDR q-value <0.25) were finally considered.

## ChIP-seq

The ChIP-seq experiment was performed twice for each antibody. $Trp53^{-/-}$ iPS cells and $Trp53^{-/-}$ iPS cells in which TRF1 expression had been abrogated (see above) were crosslinked with 1% formaldehyde (added to the medium) for 15 min at room temperature (RT). The crosslinking was stopped by adding 0.125M glycine to the medium for 5 min at RT. Fixed cells were washed with PBS containing 1 µM PMSF and protease inhibitors and then pelleted. Cells were lysed in lysis buffer (1% SDS, 10 mM EDTA and 50 mM Tris-HCl (pH 8.1)) at $2 \times 10^7$ cells/ml for 20 min at 4°C. Sonication was performed with a Covaris system (shearing time 20 min, 20% duty cycle, intensity 6,200 cycles per burst, and 30 s per cycle). 50 µg of chromatin per immunoprecipitation reaction were diluted 1/10 in dilution buffer (1% Triton X-100, 2 mM EDTA (pH 8.0), 150 mM NaCl, 20 mM Tris-HCl (pH 8.1)) and immunoprecipitated with a rabbit sera against TRF1 (raised in our laboratory), rabbit polyclonal antibody against SUZ12 (ab12073) or rabbit antibody against H3K27me3 (07–449 Upstate), using protein A/G agarose beads (Santa Cruz Biotechnology). The beads were washed once with low salt wash buffer (0.1% SDS, 1% Triton X-100, 2 mM EDTA, 20 mM Tris-HCl (pH 8.1), 150 mM NaCl), once with high-salt wash buffer (0.1% SDS, 1% Triton X-100, 2 mM EDTA, 20 mM Tris-HCl (pH 8.1), 500 mM NaCl), once with LiCl wash buffer (0.25 M LiCl, 1% NP40, 1% deoxycholateNa, 1 mM EDTA, 10 mM Tris-HCl (pH 8.1)), and twice with TE. The immune complexes were eluted with 500 µl of elution buffer (1% SDS, 0.1 M NaHCO₃). Reverse crosslinking was achieved through the addition of 20 µl of 5M NaCl and incubation at 65°C for 8 hr. DNA was recovered by RNase and proteinase K treatment, phenol/chloroform extraction and ethanol precipitation. For each sample, 10–15 ng of DNA were used for the ChIP-seq experiments. Samples were processed through subsequent enzymatic treatments of end-repair, dA-tailing, and ligation to adapters with 'NEBNext Ultra II DNA Library Prep Kit for Illumina' from New England BioLabs (catalog # E7645). Adapter-ligated libraries were completed by limited-cycle PCR and extracted with a [single] double-sided size selection for library preparation (SPRI). Resulting average fragment size is 490 bp, from which 120 bp correspond to adaptor sequences. Libraries were applied to an Illumina flow cell for cluster generation and sequenced on an Illumina NextSeq 500 sequencer, by following the manufacturer's protocols.

## CHIRT-seq and CHIRT-qPCR

CHIRT was performed once, mostly according to the protocol described by *Chu et al. (2017)* with some modifications. 30 millions cells were fixed and chromatin was fragmented in a Covaris system (shearing time 80 min, 20% duty cycle, intensity 6,200 cycles per burst, and 30 s per cycle). An anti-sense TERRA RNA biotinylated transcript consisting of eight CCCTAA repeats was used as bait for TERRA capture. After CHIRT, chromatin was eluted using RNAseH (NEB) as previously described (*Chu et al., 2017*), and the obtained DNA was used to prepare the libraries for deep sequencing. As input control, total fragmented chromatin was used. 0.5 ng of DNA per sample were used for the ChIRT-seq experiment (5 ng for the INPUT). Samples were processed through subsequent enzymatic treatments for fragmentation (5 min), end-repair, dA-tailing, and ligation to adapters with the 'NEBNext Ultra II FS DNA Library Prep Kit for Illumina' from New England BioLabs (catalog # E7805). Adapter-ligated libraries were completed by limited-cycle PCR and extracted with a [single] double-sided SPRI. Resulting average fragment size is 400 bp, from which 120 bp correspond to adaptor sequences. Libraries were applied to an Illumina flow cell for cluster generation and sequenced on an Illumina NextSeq 500 sequencer, following manufacturer's protocols. To determine changes in TERRA binding in the shTRF1 CHIRT sample compared to the control CHIRT sample, quantitative real-time PCR was performed, using the DNA libraries described above, by Go-Taq qPCR master mix (Promega) according to the manufacturer's protocol. All values were obtained in triplicates. In each case, values were normalized to input and to a genomic region where TERRA binding was not found in the CHIRT-seq. Primers are listed in *Table 1*.

## ChIP-seq and CHIRT-seq analysis

ChIP-seq and ChIRT-seq sequenced reads were processed with the RUbioSeq pipeline (*Rubio-Camarillo et al., 2017*), which performed read alignment to the mouse reference genome (NCBI37/mm9, https://ccb.jhu.edu/software/tophat/igenomes.shtml) with BWA 0.7.10 (*Li and Durbin, 2009*), removed duplicated reads with the Picard tools v1.107 (https://broadinstitute.github.io/picard/) and converted the resulting alignment files to BED format with BEDTools 2.16.2 (*Quinlan and Hall, 2010*). All ChIP and input samples were randomly normalized to the same number of reads. Peak calling was performed with MACS2 v2.0.10.20130712 (*Quinlan and Hall, 2010*). Significant peaks were associated with nearby genes using Homer v4.10.1 (*Heinz et al., 2010*) and functional annotations for associated genes were obtained with Homer and Enrichr (*Kuleshov et al., 2016*). Heatmaps with signal density over called peaks and gene TSS sites were calculated and plotted with Deeptools (*Ramírez et al., 2016*).

## Data set repository

Raw sequencing data and processed files for the RNA-seq, ChIP-seq and TERRA ChIRT-seq experiments are available from the Gene Expression Omnibus (GEO) under the ID GSE121759.

## Quantitative PCR

Samples from three independent TRF1 deletion experiments were used for validation of RNA-seq results by quantitative PCR. Total RNA from cells was extracted as described above (see RNA-seq). Reverse transcription was performed using the iSCRIPT cDNA synthesis kit (BIO-RAD) according to manufacturer's protocols. Quantitative real-time PCR was performed using Go-Taq qPCR master mix (Promega) according to the manufacturer's protocol. All values were obtained in triplicates. Primers are listed in *Table 1*. Statistical analysis were performed using one tail, unpaired Student's t-test.

## Preparation of nuclear extract, immunoprecipitation and mass spectrometry

The preparation of nuclear extract for immunoprecipitation and mass spectrometry was described in *Schneider et al. (2013)*.

## TERRA RNA-FISH

Control *Trp53*$^{-/-}$ iPS cells and *Trp53*$^{-/-}$ iPS cells depleted for TRF1 (see above) from one single experiment were seeded in 96-well bottom-glass plates (Greiner Bio-One) coated with 0.1% porcine gelatine, and grown in iPS medium (see above) supplemented with MEK inhibitor PD0325901 (1 μM)

**Table 1.** List of oligos.

| | Forward | Reverse |
|---|---|---|
| *Actin* | 5′-GGCACCACACCTTCTACAATG-3′ | 5′-GTGGTGGTGAAGCTGTAG-3′ |
| *Terf1* | 5′-TCTAAGGATAGGCCAGATGCCA-3′ | 5′-CTGAAATCTGATGGAGCACGT-3′ |
| *Myc* | 5′-GTGCTGCATGAGGAGACACC-3′ | 5′-AGGGGTTTGCCTCTTCTCC-3′ |
| *Lama1* | 5′-CTGTCACCCTGGACTTACGG-3′ | 5′-GGTTTGAACTTGACGCCATC-3′ |
| *Id1* | 5′-CTGAACGGCGAGATCAGTG-3′ | 5′-GCCTCAGCGACACAAGATG-3′ |
| *Olig2* | 5′-CAGCGAGCACCTCAAATCTA-3′ | 5′-CCCCAGGGATGATCTAAGC-3′ |
| *Foxd3* | 5′-CTTTCTTTCGGGGGACACTC-3′ | 5′-CCAGGAGCGAGCAGAGAG-3′ |
| *Zfp423* | 5′-TGACGTTCGAGAACGAGAGAG-3′ | 5′-GGGAGTCGAACATCTGGTTG-3′ |
| *Pou2f3* | 5′-ATCGACGCCAAAAGGAGAAG-3′ | 5′-TGGACAGGAGGGACTGAGAG-3′ |
| *Sgk1* | 5′-AGAGGAGTCCTGTTCCTGGG-3′ | 5′-GGTCAGGATGTTGGCATGAT-3′ |
| *Igf2* | 5′-GCTTGTTGACACGCTTCAGTT-3′ | 5′-AAGCAGCACTCTTCCACGAT-3′ |
| *Bmp4* | 5′-CGCTTCTGCAGGAACCAA-3′ | 5′-ATCAAACTAGCATGGCTCGC-3′ |
| *Hoxc12* | 5′-ACCCTGGCTCTCTGGTTTC-3′ | 5′-CAACTTCGAATACGGCTTGC-3′ |
| *Jak3* | 5′-TTGGGGACTACTTGGCTGAG-3′ | 5′-AGAAGTCCTCAGTGGCCAGA-3′ |
| *Gbx2* | 5′-CTCGCTGCTCGCTTTCTCT-3′ | 5′-GGGTCATCTTCCACCTTTGA-3′ |
| *Myd88* | 5′-TATACTGAAGGAGCTGAAGTCGC-3′ | 5′-ACACTGCTTTCCACTCTGGC-3′ |
| *Khdc3* | 5′-GAATGCCTGGAAGATCCAAA-3′ | 5′-ATGTGGGATGTGCTCTCCAT-3′ |
| *Rbm47* | 5′-AAAGAACCAGGACCAATCGC-3′ | 5′-CACTGTTGGATCGCTGTTCA-3′ |
| *Zfp345* | 5′-TGGTCTTCCCAAACATAGCC-3′ | 5′-ACTTCACGTGGGAAGAGTGG-3′ |
| *Dnmt3b* | 5′-TCTAATGCCAAAGCTCACCC-3′ | 5′-CTCTTTGCCTCTCCAAGCTG-3′ |
| *Gata2* | 5′-CACCCCTAAGCAGAGAAGCA-3′ | 5′-CAGGCATTGCACAGGTAGTG-3′ |
| *Fzd5* | 5′-AGCAGGATCCTCCGAGAGTT-3′ | 5′-CAGCACTCAGTTCCACACCA-3′ |
| *Spry2* | 5′-GATTCAAGGGAGAGGGGTTG-3′ | 5′-CTCCATCAGGTCTTGGCAGT-3′ |
| *Zscan10* | 5′-GACGGAGAGGAGGTGGTACA-3′ | 5′-GCCAAGCTCTCTTCTCTGAGG-3′ |
| *Fgf17* | 5′-ATTGATTCTCTGCTGTCAAACACA-3′ | 5′-GCTGGTATTCACGGATTTGC-3′ |
| *Sall4* | 5′-TGCCTCGGTGTTAGATGTCA-3′ | 5′-GACAAAGGTGGGCTGTGCT-3′ |
| *Klf5* | 5′-GGATCTGGAGAAGCGACGTA-3′ | 5′-TCCTCAGGTGAGCTTTTAAGTGA-3′ |
| *Nr5a2* | 5′-TGGGAAGGAAGGGACAATCT-3′ | 5′-AACGCGACTTCTGTGTGTGA-3′ |
| *Tbx3* | 5′-CATCGCCGTTACTGCCTATC-3′ | 5′-GCCAGTGTCTCGAAAACCC-3′ |
| *Spp1* | 5′-TGACCCATCTCAGAAGCAGA-3′ | 5′-CATTGGAATTGCTTGGAAGAG-3′ |
| *Smad7* | 5′-CGAATTATCTGGCCCCTGG-3′ | 5′-GACACAGTAGAGCCTCCCCA-3′ |
| *Zbtb43* | 5′-GGTAGGCTGGAGCTACGGG-3′ | 5′-TGGCCATCAAAGAGCAGTC-3′ |
| *Adm* | 5′-CATCCAGCAGCTACCCTACG-3′ | 5′-TTCGCTCTGATTGCTGGCTT-3′ |
| *Cxcl12* | 5′-CACTCCAAACTGTGCCCTTC-3′ | 5′-AATTTCGGGTCAATGCACAC-3′ |
| *Bend6* | 5′-AGAAGCATCCGGAAGGAAAA-3′ | 5′-TGCCATTCCAACCAGTTCTT-3′ |
| *Fndc5* | 5′-GGTGCTGATCATTGTTGTGGT-3′ | 5′-CCTTGTTGTTATTGGGCTCG-3′ |
| *Fzd1* | 5′-GCTTACTCCTCAGCAGCACA-3′ | 5′-TCTCTCACCCATCCGTCAGT-3′ |
| *Fzd2* | 5′-CCTCAAGGTGCCGTCCTATC-3′ | 5′-GGATCCAGAGACGGGCAAAA-3′ |
| *Jun* | 5′-ACCGAGAATTCCGTGACGAC-3′ | 5′-TGAAAAGTCGCGGTCACTCA-3′ |
| *Proser2* | 5′-ACTTGAGCAGAGGTGGCAGT-3′ | 5′-GTGCTTCAGGCTCTCGTCAT-3′ |
| *Lgi2* | 5′-CCAAGGAGTCCATCATCTGC-3′ | 5′-CATTCGGTCCTTGATTTCCA-3′ |
| *Bmp7* | 5′-CCTGGGCTTACAGCTCTCTG-3′ | 5′-CCATGAAGGGTTGCTTGTTC-3′ |
| *Pitx2* | 5′-CAAAAAGGTCGAGTTCACGG-3′ | 5′-CTTTCCTTGCTGGCCCTTAT-3′ |
| *Foxa1* | 5′-AACAGCTACTACGCGGACAC-3′ | 5′-GCTCGTGGTCATGGTGTTCA-3′ |

*Table 1 continued on next page*

*Table 1 continued*

|  | Forward | Reverse |
|---|---|---|
| *Otx2* | 5'-CTCCTGGAGGAGAGAGCAGTC-3' | 5'-GGGTCCTTGGTGGGTAGATT-3' |
| *Nkd2* | 5'-GGTGTGGAACATCGCTCAC-3' | 5'-CTAGGGAACCCTTGTCGTCC-3' |
| *Col2a1* | 5'-GCGGTCCTACGGTGTCAG-3' | 5'-TTTATACCTCTGCCCATTCTGC-3' |
| *Ppp2r2c* | 5'-GCCATCACTGATCGGAGC-3' | 5'-AGACGAAGAGGTTGCAGTGG-3' |
| *App* | 5'-CTCTACAATGTCCCTGCGGT-3' | 5'-AGCTGATTCTGGGCTCACTG-3' |
| *Satb1* | 5'-TATGAACCAGAGTTCGTTGGC-3' | 5'-TTTGCTGCTGAGACATTTGC-3' |
| *Nfix* | 5'-TCTGGCTTACTTTGTCCACACTC-3' | 5'-GTTGGGCAGTGGTTTGATGT-3' |
| *Foxn4* | 5'-ACCACTGCTCTCCACAGGAA-3' | 5'-CAGGACAGCGACTGAAGGTC-3' |
| *Trpv4* | 5'-ACCACCCCAGTGACAACAAG-3' | 5'-ATGGGCCGATTGAAGACTTT-3' |
| *Hspa1a* | 5'-TTTGTGTATTGCACGTGGGC-3' | 5'-GGGGCAGTGCTGAATTGAAG-3' |
| *Prkar2b* | 5'-AGGCTTGCAAAGACATCCTG-3' | 5'-TGTTCCCCTTCTTTGACCAAT-3' |
| *Lef1* | 5'-ACCCGTACATGTCAAATGGG-3' | 5'-GTCGCTGTAGGTGATGAGGG-3' |

| CHIRT | Forward | Reverse |
|---|---|---|
| Chr18 | 5'-CAGCCTTTGTCCTTCACAGTT-3' | 5'-GGTTCATAAGGCTTTTCTCCA-3' |
| ChrX | 5'-TGTTCCCTCACAGCACAGAG-3' | 5'-TAAGCCAGCCTCTCCAAAGA-3' |
| *Pros1* | 5'-GGCAGTCTCTGGAGTTGGAA-3' | 5'-CTAGCATCCCTTCCCCATTC-3' |
| *Ddx58* | 5'-AAGTGGGGTTTCAGAGAGCA-3' | 5'-CCCTAACCCTTCCCCATAAA-3' |
| *Arid5b* | 5'-CTCTTCCCCTGGAGATCCTT-3' | 5'-TTGGAAACAGATTTGAGCATTC-3' |
| *Syde2* | 5'-GCTGGGTTTACCCCAATACA-3' | 5'-GACCCACTTCCTAAGGACAGAA-3' |
| *Zdbf2* | 5'-CATGGGGAAAGCATAATTGC-3' | 5'-AGGCTTGGGACTCCTCTTGT-3' |
| *Olig2* | 5'-CCACACCCTGTGTGTCTGTC-3' | 5'-TCAACCTTCCGAACTTGAGG-3' |
| *Lrrc8c* | 5'-GCTAGGTTCTGGGGACTGG-3' | 5'-CAACCGCGTTTTCTCCTAGT-3' |
| *Sorl1* | 5'-GCCTACCTCAGAATGGAGGTC-3' | 5'-CACACACACACACTACCATATAATCC-3' |
| *Mpdz* | 5'-GCGTCCCATCTTAAAACCAA-3' | 5'-GATCCTCTCCATCCCTACCC-3' |
| *Prmt8* | 5'-GGTTTGGGACTTAGGGGAAC-3' | 5'-AGTTCCTTTCCCCCTTGAAA-3' |
| *Dleu7* | 5'-TCAAGACTGGACCCCAAAAC-3' | 5'-GGACCAGCCAGCTTGTATGT-3' |
| *Dcc* | 5'-TTCAGTCCCTGGACAGACAG-3' | 5'-ACACGCCTTTCCTTCACAGT-3' |
| Control region | 5'-CAATGCCTAGATATACCGATCTCTT-3' | 5'-CTCAGGACAAGACCCCACTG-3' |

DOI: https://doi.org/10.7554/eLife.44656.022

and GSK3 inhibitor CHR99021 (3 µM). Next day, the cells were washed with PBS twice and incubated in cytobuffer (100 mM NaCl, 300 mM sucrose, 3 mM MgCl$_2$, 10 mM pipes (pH 6.8)) for 30 s, incubated in cytobuffer with 0.5% Triton X-100 for 30 s, incubated in cytobuffer for 30 s, and fixed for 10 min in 4% paraformaldehyde in PBS. Fixed cells were dehydrated in 70% ethanol three times, once in 80%, 95%, and 100% ethanol, air-dried, and hybridized overnight at 37°C with a telomere-specific PNA-FITC probe (Panagene) in hybridization buffer (2 × sodium saline citrate (SSC)/50% formamide). Next day, cells were washed twice for 15 min in 2 x SSC, 50% formamide at 40°C, twice for 10 min in 2 × SSC at 40°C, for 10 min in 1 x SSC at 40°C, for 5 min in 4 x SSC at room temperature and for 5 min in 4 x SSC containing 0.1% Tween-20. The cells were incubated with DAPI (Molecular Probes) at room temperature for 10 min and washed three times with PBS. Fluorescence signal was preserved in Vectashild (Vector laboratories). Signals were visualized in a confocal ultra spectral microscope SP5-WLL (Leica). Statistical analyses were performed using Student's t-test, where n indicates the number of cells of each sample analyzed.

## TERRA northern blot

Northern blot analyses were performed using standard protocols. Telomere probe was obtained by excising a 1.6 Kb (TTAGGG)$_n$ DNA insert from pNYH3 plasmid (a kind gift from T. de Lange, Rockefeller University, NY, USA). Northern blots were normalized using 18S or 28 s probes and quantified using ImageJ. The probes were labeled using the commercial Prime-It II Random Primer Labeling Kit (Agilent Genomics). For TRF1 deletion in iPS cells, samples were obtained from three independent TRF1 deletion experiments. Statistical analysis was performed using Student's t-test. For TERRA analysis in primary keratinocytes from newborn mice, three newborn mice from each genotype were used. Statistical analysis was performed using Student's t-test.

## Immunofluorescence

2i-grown *Trp53*$^{-/-}$ iPS cells and 2i-grown *Trp53*$^{-/-}$ iPS cells where TRF1 expression had been abrogated, or ES cells treated or untreated with retinoic acid, were plated in gelatin-coated coverslips, incubated with cytobuffer (100 mM NaCl, 300 mM sucrose, 3 mM MgCl$_2$, 10 mM pipes (pH 6.8)) with 0.5% Triton X-100 for 6 min, and fixed for 10 min in 4% paraformaldehyde in PBS. Cells were then blocked in 10% BSA in PBS for 1 hr at RT and incubated with the primary antibodies dissolved in Dako antibody diluents (Dako) for 1 hr in a humid chamber at RT. The following antibodies were used: rat antibody anti-TRF1 (home made, 1:250), rabbit antibody against RAP1 (A300-306A, Bethyl, 1:250) and mouse monoclonal antibody against phospho histone H2A.X (Ser139) (Millipore 05–636, 1:250). Cells were washed 3 times for 30 min with PBS containing 0.1% Tween-20 and incubated with Alexa secondary antibodies for 1 hr at RT in a humid chamber. After 3 washes of 30 min with PBS containing 0.1% Tween-20, samples were mounted in Prolong with DAPI. Fluorescent signals were visualized in a confocal ultraspectral microscope SP5-WLL (Leica). The experiment was performed once. Statistical analysis were performed using Student's t-test, where n indicates the number of cells in each analyzed sample.

## Enrichr analysis

The p-values shown in all the figures containing Enrichr analysis (Wikipathways, KEGG Pathways, ENCODE histone modifications, CHEA) were the adjusted p-values provided by Enrichr.

## RNA immunoprecipitation (RIP)

*Trp53*$^{-/-}$ 2i-grown iPS cells, either controls or cells in which TRF1 was depleted) were crosslinked with 1% formaldehyde for 15 min and the quenched with glycine 125 mM for 5 min. Cells were washed with PBS, scraped from the plate and pelleted for 5 min at 1200 rpm. $25 \times 10^6$ cells per immunoprecipitation were resuspended in 7 ml of ice-cold cell lysis buffer (5 mM Tris (pH 8), 85 mM KCl, 0.5% Igepal), and incubated for 10 min at 4°C in rotation. Crude nuclear sample was collected by centrifugation for 10 min at 3000 g at 4°C. The nuclear pellet was washed with 0.5 ml of cell lysis buffer without Igepal (5 mM Tris (pH 8), 85 mM KCl) and microcentrifuged for 10 min at 3000 g at 4°C. The nuclear pellet was resuspended in 1 ml of RIPA buffer (150 mM NaCl, 50 mM Tris (pH 7.4), 0.1% SDS, 0.5% Na-Doc, 1% Triton +0.1M DTT, RNAsin, Proteinase inhibitor 1x) to lyse the nucleus and incubated in ice for 10 min. The sample was doused 20 times and sonicated thrice for 5 s on setting 3. The sonicated lysate was microcentrifuged for 10 min at 4°C at 14,000 rpm. A fraction of the soluble chromatin (supernatant) was kept as input. The sample was pre-cleared with Protein A+G beads for 1 hr at 4°C and centrifuged at 2000 rpm for 4 min. The precleared sample was then incubated o/n with the corresponding antibody (IgG as negative control or antibody anti SUZ12 (ab12073)). After o/n incubation, the beads were washed once with low-salt buffer (0.1% SDS, 1% Triton, 2 mM EDTA, 20 mM Tris (pH 8), 150 mM NaCl, +0.1M DTT, RNAsin, Proteinase inhibitor 1x), once with high-salt buffer (0.1% SDS, 1% Triton, 2 mM EDTA, 20 mM Tris (pH 8), 500 mM NaCl, +0.1M DTT, RNAsin, Proteinase inhibitor 1x), once with LiCl buffer (0.25 M LiCl, 1% Igepal, 1% deoxycholate, 1 mM EDTA, 10 mM Tris (pH 8), +0.1M DTT, RNAsin, Proteinase inhibitor 1x), and twice with TE (pH 8) (100 mM Tris (pH 8), 10 mM EDTA (pH 8)). Each wash was performed at 4°C for 5 min in rotation. The RNA was eluted with 250 ul of elution buffer (25 mM Tris (pH 7.5), 5 mM EDTA, 0.5% SDS), treated with proteinase K for 45 min at 45°C and incubated 2 hr at 65°C to reverse crosslink. RNA extracted with trizol and purified and DNAse-treated on using RNeasy Kit (QUIAGEN 74106),

following the manufacturer's instructions. The RNA was loaded using a dot-blot and the presence of TERRA and 18S RNAs was detected as described before (*Marion et al., 2009*).

## Acknowledgements

We thank Ana Cuadrado for her help with the setting up of ChIP-seq. We thank Tommaso Vicanolo for his help in differentiating cells. We are indebted to Orlando Domínguez and the Genomic Unit from CNIO for their advice and for the generation of the libraries. We thank Diego Megías from the Confocal Microscopy Unit at CNIO for his help with the quantification of the immunofluorescence and RNA-FISH. We are grateful to Jeannie T Lee and Catherine Hsueh-Ping Chu for kindly sharing the CHIRT protocol and for assistance with details. Research in the Blasco lab is funded by the Spanish Ministry of Economy and Competitiveness Projects (SAF2013-45111-R and SAF2015-72455-EXP), the Comunidad de Madrid Project (S2017/BMD-3770), the World Cancer Research (WCR) Project (16–1177) and the Fundación Botín (Spain).

## Additional information

### Competing interests

José Alejandro Palacios-Fábrega: The author is affiliated with Astellas Pharma Europe Ltd. The author has no other competing interests to declare. The other authors declare that no competing interests exist.

### Funding

| Funder | Grant reference number | Author |
| --- | --- | --- |
| Ministerio de Economía y Competitividad | SAF2013-45111-R | Isabel López de Silanes |
| Consejería de Educación, Juventud y Deporte, Comunidad de Madrid | S2017/BMD-3770 | Juan J Montero<br>Maria A Blasco |
| World Cancer Research Fund | 16-1177 | Maria A Blasco |
| Fundación Botín | | Maria A Blasco |
| Ministerio de Economía y Competitividad | SAF2015-72455-EXP | Maria A Blasco |

The funders had no role in study design, data collection and interpretation, or the decision to submit the work for publication.

### Author contributions

Rosa María Marión, Formal analysis, Designed, performed and interpreted most of the experiments, Writing – original draft, Writing – review and editing; Juan J Montero, Designed, performed and interpreted most of the experiments; Isabel López de Silanes, Performed the differentiation experiment; Osvaldo Graña-Castro, Performed the bioinformatic analysis; Paula Martínez, Stefan Schoeftner, Provided the keratinocytes, Performed Northern Blots; José Alejandro Palacios-Fábrega, Performed the proteomics analysis; Maria A Blasco, Secured funding, Conceived the original idea, Designed and interpreted results, Writing – original draft, Writing – review and editing

### Author ORCIDs

Rosa María Marión (iD) https://orcid.org/0000-0002-0374-3884
Maria A Blasco (iD) https://orcid.org/0000-0002-4211-233X

### Decision letter and Author response

Decision letter https://doi.org/10.7554/eLife.44656.030
Author response https://doi.org/10.7554/eLife.44656.031

## Additional files

### Supplementary files

• Supplementary file 1. Table S1.
DOI: https://doi.org/10.7554/eLife.44656.023

• Supplementary file 2. Table S2.
DOI: https://doi.org/10.7554/eLife.44656.024

• Supplementary file 3. Table S3.
DOI: https://doi.org/10.7554/eLife.44656.025

• Supplementary file 4. TRF1 does not interact with SUZ12. Mass spectroscopy of protein complexes immunoprecipitated with an anti-GFP antibody in $Terf1^{GFP/GFP}$, $Terf1^{+/GFP}$ and $Terf1^{+/+}$ iPS cells. The table shows the 64 proteins that showed co-immunoprecipitation in both $Terf1^{GFP/GFP}$ and $Terf1^{+/GFP}$ cells but not in $Terf1^{+/+}$ cells. Note the presence in this group of proteins of different components of shelterin, namely TPP1 (ACD), RAP1 (TERF2IP), TRF1 (TERF1), TIN2 (TINF2) and POT1, but not the presence of SUZ12. Related to *Figure 2*.
DOI: https://doi.org/10.7554/eLife.44656.010

• Transparent reporting form
DOI: https://doi.org/10.7554/eLife.44656.026

### Data availability

Raw sequencing data and additional processed files for the RNA-seq, ChIP-seq and TERRA CHIRT-seq experiments have been placed in the Gene Expression Omnibus (GEO) under the accession number GSE121759.

The following dataset was generated:

| Author(s) | Year | Dataset title | Dataset URL | Database and Identifier |
|---|---|---|---|---|
| Marión RM, Montero JJ, Graña-Castro O, Blasco MA | 2018 | RNA-seq, ChIP-seq and TERRA CHIRT-seq from p53-/- iPS infected with a lentiviral virus carrying a control scrambled shRNA or shRNA against TRF1 | https://www.ncbi.nlm.nih.gov/geo/query/acc.cgi?&acc=GSE121759 | NCBI Gene Expression Omnibus, GSE121759 |

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
