## [Decision Letter]

Thank you for submitting your article "TERRA regulate the transcriptional landscape of pluripotent cells through TRF1-dependent recruitment of PRC2" for consideration by *eLife*. Your article has been reviewed by three peer reviewers, one of whom is a member of our Board of Reviewing Editors, and the evaluation has been overseen by Jessica Tyler as the Senior Editor. The following individual involved in review of your submission has agreed to reveal her identity: Hsueh Ping Chu (Reviewer #3).

The reviewers have discussed the reviews with one another and the Reviewing Editor has drafted this decision to help you prepare a revised submission.

Summary:

Over the past decade, the Blasco lab has published a series of papers regarding on the telomeric repeat-containing RNA (TERRA) and the roles of TRF1 in cancer and pluripotent cells. As a continuation of her research, Marión et al., studied the impact of TRF1 on maintaining pluripotency in iPS cells. TERRA level increased as a result of TRF1 depletion and, under such situation, TERRA was recruited to the genes for stem cell renewal and embryonic development. Authors also conducted a series of ChIP-seq experiments, for which demonstrating PRC2 component SUZ12 and H3K27me3 histone mark enrichment on pluripotency and differentiation genes. Expression level was significantly changed among those genes showing concomitant SUZ12 and H3K27me3 enrichment, as they are recruited to the transcription start sites (TSS). SUZ12 and H3K27me3 enriched genes are also related to pluripotency and development, respectively, as shown in RNA-seq assays. TERRA CHIRT peaks were observed in the TSS of genes having altered expression, in certain case overlapped with SUZ12 and H3K27me3 association. TERRA bound genes are also often related to pluripotency and development. Additionally, TRF1 depletion may regulate gene expression via direct binding, authors revealed that TRF1 depletion cause Laminin A down-regulation which have TRF1 binding site in the intron. However, extra-telomeric TRF1 binding is only restricted on a few gene loci, would not suffice to explain the majority of epigenetic changes in TRF1 depleted cells. Overall, authors claim that TRF1 is essential to maintain pluripotency in stem cells, by suppressing TERRA expression and subsequent genome-wide PRC2 localization, thereby stabilizing epigenetic marks on pluripotency genes of stem cells. This paper contains potentially interesting and important data. The study is potentially valuable, but a number of technical issues need to be addressed before further consideration.

Essential revisions:

Reviewer 1: See points 1-3 and point 5 in the detailed review

Reviewer 2: A number of substantial concerns were raised by this reviewer. It seems important to us to deplete TERRA using gapmer ASO in TRF1-depleted cells with RNA-Seq or RT-qPCR of selected genes. The reviewer also raised the fact that TERRA upregulation in TRF1-depleted cells showed no changes in TERRA levels in TRF1-deleted MEFs by the de Lange lab. This contradiction needs to be explained. Finally, the Northern blots in Figure 5 remain problematic (but with thanks for the earlier explanation provided). The reviewer specifically noted that the TERRA pattern is different in each panel, there is no TERRA signal in panel C for MEFs, while mouse TERRA is definitely abundant and easily detectable. Similarly, there is no signal for TERRA in wt cells in panel D. Why does the 18S signal fade away till disappearing throughout samples in panel C?

Reviewer 3: This reviewer was generally supportive but would also like to see some additional experiments. Specifically, they would like evidence showing the TERRA mediates the increased recruitment of SUZ12 after TRF1 depletion. A RIP experiment after depletion seems appropriate. A good question is also about why SUZ12 and TERRA bind at different sites if TERRA is recruiting PRC2. Secondly, biological replicates for ChIP-seq data are critical. Finally, various contradictions mentioned by this reviewer (as well as the other reviewers) need to be addressed directly.

Reviewer #1:

This paper contains a large body of meta-analysis data, which should be useful for researchers studying on telomeres, long noncoding RNAs, epigenetic regulations, and stem cell biology. The authors also reveal a novel function of shelterin component TRF1, as one of the genes for maintaining stem cell identity by impeding TERRA promoted genome-wide epigenetic changes. Also, this paper connects the epigenetic roles of TERRA via interacting PRC2 (Chu et al., 2017; Montero et al., 2018; Wang et al., 2017) with TRF1 upregulation as a stem cell marker (Schneider et al., 2013). The study is therefore potentially valuable, but a number of technical issues need to be addressed before publication.

1) Despite efficient TRF1 depletion, telomeres are not recognized as γH2AX TIF (Figure 1—figure supplement 1). This data directedly contradicts previous TRF1 studies including Blasco lab's publication on TRF1 knock-out iPS cells (Schneider et al., 2013). Cells might be harvested earlier than the moment of DDR activation (as written on the manuscript), however, authors need to provide the information that whether shTRF1 telomeres are inflicted with DNA damage responses as described before.

2) Overall links between the events of TERRA binding, PRC2 localization, and subsequent gene regulation are fairly weak. As shown in Figure 7D and E, only a fraction of TERRA bound genes shows both SUZ12 and H3K27me3 enrichment (13 / 101 of down-regulated genes and 25 / 142 of up-regulated genes). Furthermore, the SUZ12 and/or H3K27me3 increase in shTRF1 cells are not quite convincing (Figure 7D and E; Figure 3B, H3K27me3 peaks; Figure 4B, H3K27me3 peaks). TERRA protein interatom (Chu et al., 2017) covers a list of epigenetic regulators and transcription factors, therefore, TERRA mediated transcription regulations do not necessarily undergo PRC2 recruitment. Although TRF1 depletion convincingly increased SUZ12 level in certain genes (Figure 2 and 3), the large part of gene expression changes happened in genes without TERRA accumulation (Figure 7A). Therefore, most transcriptional changes are due to the systemic effect of TRF1 depletion rather than TERRA accumulation in such genes. Nevertheless, a significant portion of altered gene expression in shTRF1 cells coincide with TERRA bound genes (Figure 7A, 98/328 of down-regulated genes and 162/483 of up-regulated genes) and PRC2 (SUZ12) accumulation was clearly observed at pluripotency and differentiation genes. This study does not sufficiently support a model in Figure 8 as also claimed in the manuscript, but still can deliver important information of TRF1 functions on stem cells.

3) H3K27me3 is an epigenetic mark tightly associated with gene suppression. However, this study demonstrated that a number of genes are upregulated with increased H3K27me3. Authors need to provide a sufficient explanation of these seemingly contradictory results.

4) Physiological information is somewhat missing in this study. Authors can enhance their point on the TRF1 function by measuring TRF1 and TERRA level changes during early cell differentiation.

5) TRF1 is not a sole shelterin component regulating TERRA, previous studies show greater TERRA upregulation upon TRF2 depletion (Porro et al., 2014; Rossiello et al., 2017). Authors may make clear the insight on the TERRA mediated gene regulations with TRF2 depleted p53 -/- cells.

Reviewer #2:

This is a potentially interesting paper suggesting a possible role for TRF1 in regulating pluripotency through TERRA expression and binding to extra-telomeric chromosomal loci. The authors show that 1) TRF1 depletion in 2i grown iPS cells leads to alteration of expression of genes involved in maintaining pluripotency and redistribution of the PRC2 component SUZ12 and of the heterochromatin mark H3K27me3; 2) TRF1 binds to a few extratelomeric loci in iPS cells, although this event does not seem to be directly responsible for the regulation of pluripotency-associated genes; 3) TRF1 depletion in different cells (including iPS) increases TERRA levels; 4) TRF1 depletion induces TERRA binding to extrachromosomal loci including a large number of genes deregulated when TRF1 is depleted.

The data are interesting as they might indeed explain why TRF1 is involved in pluripotency maintenance, yet the study remains largely descriptive/correlative. While the authors' model makes sense with the data, some validation experiments are necessary to validate the proposed mechanisms. Depleting TERRA in TRF1-depleted cells followed by RNA-Seq or RT-qPCR of selected genes would confirm of disprove the model. Anti-TERRA gapmers have been used successfully in mouse cells, thus this experimental plan should not pose major difficulties.

TERRA upregulation in TRF1-depleted cells is somehow surprising as the de Lange group showed no changes in TERRA levels in TRF1-deleted MEFs. Could this be linked to the p53 status of the used cells? That specific paper needs to be referenced as it was the first one describing a TRF1 KO mouse. Moreover, the Northern blots shown in Figure 5B-D are not satisfactory. TERRA pattern is different in each panel, there is no TERRA signal in panel C for MEFs, while mouse TERRA is definitely abundant and easily detectable. Similarly, there is no signal for TERRA in wt cells in panel D. Why does the 18S signal fade away till disappearing throughout samples in panel C?

It would be important to test whether the increased in TERRA depends on transcription.

*Reviewer #3:*

In this manuscript, the authors (Marion, et al.) investigate the function of TRF1 in pluripotent cells and demonstrate that the TRF1 removal leads to the changes of gene expression that are related to the pluripotency. They also found that the enrichment of SUZ12 and H3K9me3 were increased in genes that are involved in the development and cell differentiation. In the end, they demonstrated that a long non-coding RNA, TERRA, is induced upon TRF1 depletion and the increase of TERRA binding to genome correlates the patterns of SUZ12 and H3K9me3 enrichment. The authors propose a model that TERRA mediates the recruitment of PRC2 complex in TRF1 depleted cells. Their findings are exciting and provide a mechanism by which TRF1 not only functions at telomeres but also is essential for the epigenetic regulation in ES cells. However, there are a few concerns that should be addressed that could make the manuscript more convincing and more interesting for the readership of *eLife*.

1) There is no direct evidence showing the TERRA mediates the increased recruitment of SUZ12 to those genomic loci after TRF depletion in this study. Does TRF1 removal increase the interaction between PRC2 and TERRA in vivo? This is an important question that could greatly support the author's claim. RNA immunoprecipitation (such as UV-RIP) for PRC2 complex could be done to address this question. Figure 7D and 7E show that the binding sites of SUZ12 and TERRA are nearby but not at the same sites. If TERRA mediates the PRC2 recruitment, why SUZ12 and TERRA bind at different sites?

2) I couldn't find anywhere that the authors mentioned the biological replicates for ChIP-seq data. If not, it is crucial to have at least two biological replicates of ChIP-seq or ChIP-qPCR to confirm the increase of SUZ12 and H3K27me3 enrichment at several genomic loci after TRF1 depletion.

3) The title seems to contradict the author's results in the manuscript. TERRA recruits more PRC2 complex to the genomic loci in the absence of TRF1. So it means that TERRA can regulate the transcriptional landscape of pluripotent cells through a TRF1-independent manner (not dependent). In the fourth paragraph of the Discussion, the authors also state that TERRA can also bind to PRC2 sites in the genome that are dependent on TRF1. If TERRA binding to PRC2 sites is in a TRF1-dependent manner, the results should show the decrease of TERRA binding to PRC2 sites upon TRF depletion.

[Editors' note: further revisions were requested prior to acceptance, as described below.]

Thank you for resubmitting your work entitled "TERRA regulate the transcriptional landscape of pluripotent cells through TRF1-dependent recruitment of PRC2" for further consideration at *eLife*. Your revised article has been further evaluated by Jessica Tyler as the Senior Editor, a Reviewing Editor, and two reviewers.

The manuscript has been improved but there are some remaining, significant issues that need to be addressed before acceptance, as outlined below:

Reviewer #2:

The paper has now improved, yet some main issues remain.

1) The TERRA depletion experiment is not satisfactory according to me. First of all, a 10/20% depletion efficiency is essentially no depletion. With such depletion efficiencies I would argue that any observed cellular responses derive from introducing a C-rich telomeric RNA sequence in cells, rather than from depletion of TERRA. Additionally, measuring TERRA depletion using FISH is not ideal as the signal might diminish at least in part because the C-rich gapmer is preventing the FISH probe from binding, possibly in absence of TERRA degradation in cells. A Northern blot should be performed. Also, how does the TERRA signal look like? There is no image shown, as to give a sense of how this minor depletion can be visualized. A complementation experiment where TERRA is ectopically re-expressed is formally impossible using the telomeric C-rich gapmers, as the telomeric TERRA sequence cannot be changed. Nevertheless, the authors could also consider to deplete TERRA using one or more gamers targeting unique sequences from the subtelomere of chromosome 18, which is the TERRA locus according to results from the Blasco lab. In this case, a complementation experiment could be performed by expressing ectopic telomeric RNA, which should localize in trans to all telomeres, again according to what published by the Blasco lab. Issues stem also from the fact that siRNA-mediated depletion of TRF1 is very poor (at least at the mRNA levels; I am not sure I understand why the authors did not perform a Western blot). Why are shRNA and gapmers 'incompatible'? Overall, as they stand now, I do not think that those experiments should be published.

2) I still think that the quality of the Northern blots shown in Figure 5 is quite low. For panel D (which did not change compared to the last version), I suggest using the images shown in Author response image 1, they are definitely better.

3) The authors now back up their finding in mouse cells by citing several reports showing TERRA up-regulation in TRF1-depleted human cells of various type. This is ok and possibly interesting in terms of evolutionary conservation of this mechanism. Unfortunately, the authors forgot to cite their own paper appeared in Nat Cell Biol in 2008 (Schoeftner and Blasco, 2008), where siRNA-mediated depletion of TRF1 in C2C12 mouse cells was shown to decrease TERRA levels, and TRF1 was thus presented as a negative regulator of TERRA transcription through its interaction with RNA polymerase II. This should also be explained.

Reviewer #3:

The revised version of the manuscript provides some additional work such as RNA-seq and RNA-IP experiments to strengthen the authors' idea that TRF1 regulating gene expression in pluripotent cells is mediated by TERRA and SUZ12 recruitment to the genome. They found that the interaction of TERRA and SUZ12 was increased after TRF1 depletion. They also used siRNA and LNA Gapmer to knockdown TRF1 and TERRA in 2i-grown iPS cells and investigated the transcriptome changes between TRF1-alone depletion and TRF1/TERRA depletion. They show that the expression of some genes, which were altered in TRF1-alone depletion, was not altered in TRF1/TERRA depletion, suggesting that the upregulation of TERRA levels is required for the changes in gene expression upon TRF1 depletion. However, the authors didn't compare the results with the gene targets for the increase of TERRA binding and SUZ12 binding after TRF1 depletion. Therefore, it is not clear whether the recruitment of TERRA and SUZ12 is required for the regulation of those genes. This question needs to be addressed before publication.

1) In Figure 7—figure supplement 1C, the authors show that the expression of some genes that were not altered when downregulation of TERRA after TRF1 depletion (such as Lef1 and Stk39), indicating that TRF1 regulates the expression of these genes by regulating TERRA levels. Do these genes have TERRA binding sites and SUZ12 binding sites? And do they correlate with the increase of TERRA binding and SUZ12 binding after TRF1 depletion? These questions are important to rule out whether TERRA regulates the gene expression is mediated by recruiting SUZ12 or other mechanisms. In Figure 7J, the authors also show that up or down-regulated genes after TRF1 depletion have TERRA binding sites nearby. Does the expression of those genes change in both TRF1 and TERRA depleted cells? Because I don't see any genes are presented in both Figure 7—figure supplement 1C and Figure 7J, the connection between TRF1 depletion, the increase of TERRA binding sites, and the increased of SUZ12 binding sites is weak. It would be more convincing that the authors show the gene expression of some specific genes (up or down-regulated in shTRF1 cells having TERRA and SUZ12 binding sites), which are controlled by TERRA and SUZ12 in a TRF1 dependent manner (TRF1 expression). In other words, the authors should find some common genes in Figure 7J and Figure 7—figure supplement 1C. The authors could check TERRA CHIRT qPCR for those genes found in Figure 7—figure supplement 1C using shTRF1 cells.

2) The CHIRT-seq technique can detect RNA interacting chromatin that could be either mediated by protein complexes or through direct RNA/DNA interaction. The sentence claiming that "The CHIRT technique can only detect peaks where RNA-DNA interaction occurs." in the main text is incorrect and misleading. CHIRT-seq (Chu et al., 2017) utilizes glutaraldehyde to fix cells, and this reagent will crosslink the protein-protein complex, protein-RNA, protein-DNA complex, and RNA-protein-DNA complex. Other methods such as ChIRP-seq (Chu et al., 2012) developed by Howard Chang's laboratory and CHART-seq (Simon et al., 2013) developed by Robert Kingston's laboratory also use aldehyde-based chemicals to crosslink RNA-chromatin complex. The major purpose of these techniques is to detect RNA associated complexes that are made of RNA, proteins, and DNA. The authors need a better explanation of why SUZ12 and TERRA have different binding sites. One possibility is that ChIP and CHIRT use physical sonication to shear chromatin, and thus the long-range interaction mediated by long non-coding RNA could be disrupted after sonication. Other mechanisms that may lead to this result should also be discussed in the Discussion section.

---

## [Author Response]

Essential revisions:Reviewer 1: See points 1-3 and point 5 in the detailed review

We have addressed all concerns raised by this reviewer (see also “detailed answer to reviewer #1). In particular:

As requested by the reviewer #1, we now show that a type of telomere aberration previously associated to TRF1 dysfunction, namely the so-called “multitelomeric signals or MTS”, are significantly increased as the consequence of TRF1 downregulation (see new Figure 1—figure supplement 1A), thus supporting lower TRF1 levels at telomeres.

As also requested by reviewer #1, we have now depleted TRF2 in iPS cells (see new Figure 5—figure supplement 1A) and analyzed the levels of TERRA (see new Figure 5—figure supplement 1B-C). We did not find any changes in TERRA expression in TRF2-downregulated iPS cells, showing the specificity of TRF1 depletion in altering TERRA levels in iPS cells (see new Figure 5—figure supplement 1B-C).

Finally, in the revised manuscript text, we now explain the reasons why TERRA peaks and SUZ12 peaks are not totally overlapping (see subsection “Genomic TERRA binding correlates with genes that are differentially expressed in the absence of TRF1 and that have presence of SUZ12 and H3K27me3”, third paragraph), as well as we clarify the reason why some genes upregulated upon TRF1 depletion show increased H3K27me3 signal (see subsection “Depletion of TRF1 induces recruitment of SUZ12 and H3K27me3 to genes de-regulated in the absence of TRF1”, second paragraph).

Reviewer 2: A number of substantial concerns were raised by this reviewer. It seems important to us to deplete TERRA using gapmer ASO in TRF1-depleted cells with RNA-Seq or RT-qPCR of selected genes. The reviewer also raised the fact that TERRA upregulation in TRF1-depleted cells showed no changes in TERRA levels in TRF1-deleted MEFs by the de Lange lab. This contradiction needs to be explained. Finally, the Northern blots in Figure 5 remain problematic (but with thanks for the earlier explanation provided). The reviewer specifically noted that the TERRA pattern is different in each panel, there is no TERRA signal in panel C for MEFs, while mouse TERRA is definitely abundant and easily detectable. Similarly, there is no signal for TERRA in wt cells in panel D. Why does the 18S signal fade away till disappearing throughout samples in panel C?

We have fully addressed these concerns reviewer #2 (see also “detailed answer to reviewer #2”) in the revised manuscript. In particular:

As requested by reviewer #2, in the revised manuscript we have now downregulated TERRA in TRF1-depleted cells by using Gapmers (see Figure 7—figure supplement 1 and analyzed the impact in gene expression (see Figure 7—figure supplement 1). Due to technical incompatibilities with the use of shRNAs and Gapmers, we used siRNAs to downregulate TRF1 levels and Gapmers to decrease TERRA expression. Thus, 2i-grown *p53^-/-^*iPS cells were co-transfected with a combination of two different siRNAs against TRF1 and a Gapmer against TERRA. In this experimental setting, we obtained up to 25-30% downregulation of TERRA levels in these cells by using Gapmers (see Figure 7—figure supplement 1A) and a moderate decrease in TRF1 levels (see Figure 7—figure supplement 1B). In spite of the technical difficulties of the experiment and the moderate downregulation of TRF1 and TERRA, we went ahead and performed RNA-seq of the samples. We observed that a number of genes that we had found altered when depleting TRF1 by shRNA were also regulated when depleting TRF1 with an siRNA (see Figure 7—figure supplement 1C), although the changes were modest, most likely owing to the modest downregulation of TRF1. Importantly, we now show that expression of some of these genes was not altered when downregulation of TERRA was performed at the same time that depletion of TRF1 (see Figure 7—figure supplement 1C), indicating that their expression change depends on TERRA upregulation. Interestingly, ENRICH analysis of the downregulated genes showed that they are targets of pluripotency factors and belong to the PluriNetWork pathway (see Figure 7—figure supplement 1D), while upregulated genes are targets of pluripotency factors, differentiation factors and Polycomb, and belong to differentiation pathways (see Figure 7—figure supplement 1E). Therefore, these results are in agreement with our model, in which the upregulation of TERRA levels is required for the changes in gene expression produced upon TRF1 depletion.

In the revised manuscript, we now discuss the apparent discrepancy with the de Lange results (see subsection “Depletion of TRF1 in 2i-grown iPS cells induces the up-regulation of TERRA RNAs expression”, first paragraph). Finally, we repeated the Northern blot shown in former Figure 5C and we now clearly show that TERRA increases associated to TRF1 downregulation (see new Figure 5C).

Reviewer 3: This reviewer was generally supportive but would also like to see some additional experiments. Specifically, they would like evidence showing the TERRA mediates the increased recruitment of SUZ12 after TRF1 depletion. A RIP experiment after depletion seems appropriate. A good question is also about why SUZ12 and TERRA bind at different sites if TERRA is recruiting PRC2. Secondly, biological replicates for ChIP-seq data are critical. Finally, various contradictions mentioned by this reviewer (as well as the other reviewers) need to be addressed directly.

We have addressed the concerns reviewer #2 (see also “detailed answer to reviewer #2”) in the revised manuscript. In particular:

As requested by reviewer #3, we now include in the revised manuscript a RIP experiment after depletion of TRF1 and show that the binding of SUZ12 to TERRA clearly increases after TRF1 depletion (see new Figure 8A). Also as requested by this reviewer, in the revised manuscript we now include a second biological replicate of all the ChIP-seq (see new Figure 2—figure supplement 1 and Figure 2—figure supplement 4). In particular, we find that heatmaps of the two biological replicates of H3K27me3 ChIP-seq, with reads plotted around the peaks for both replicas, show the same pattern (see new Figure 2—figure supplement 1). Cluster 3 and 7 include regions when H3K27me3 increases upon TRF1 depletion in both replicas. Importantly, more than 90% of the genes that show increased levels of H3K27me3 mark upon TRF1 depletion are the same in both replicas, confirming the high similarity between both ChIP-seq.

Finally, we have addressed all the other suggestions of this reviewer in the revised manuscript text and figures.

Reviewer #1:[…] 1) Despite efficient TRF1 depletion, telomeres are not recognized as γH2AX TIF (Figure 1—figure supplement 1). This data directedly contradicts previous TRF1 studies including Blasco lab's publication on TRF1 knock-out iPS cells (Schneider et al., 2013). Cells might be harvested earlier than the moment of DDR activation (as written on the manuscript), however, authors need to provide the information that whether shTRF1 telomeres are inflicted with DNA damage responses as described before.

The reviewer makes a good point, however, this can be explained because we have used very short time points after TRF1 deletion precisely not to complicate the interpretation of our results by the activation of a DNA damage response (DDR) as a consequence of TRF1 abrogation. Nevertheless, in the revised manuscript, we now included the analysis of telomere aberrations previously associated to TRF1 abrogation, such as the so-called “multitelomeric signals” or MTS. To this end, we performed quantitative telomere FISH (QFISH) in metaphases obtained after TRF1 depletion in p53^-/-^iPS cells (see new Figure 1—figure supplement 1A). The new results demonstrate that TRF1 downregulation increases the abundance of “multitelomeric signals”, as expected from TRF1 depletion, however, the short time of cell harvesting after TRF1 depletion is probably not sufficient for the activation of a DDR over the basal levels observed in the p53-deficient control cell lines.

2) Overall links between the events of TERRA binding, PRC2 localization, and subsequent gene regulation are fairly weak. As shown in Figure 7D and E, only a fraction of TERRA bound genes shows both SUZ12 and H3K27me3 enrichment (13 / 101 of down-regulated genes and 25 / 142 of up-regulated genes). Furthermore, the SUZ12 and/or H3K27me3 increase in shTRF1 cells are not quite convincing (Figure 7D and E; Figure 3B, H3K27me3 peaks; Figure 4B, H3K27me3 peaks). TERRA protein interatom (Chu et al., 2017) covers a list of epigenetic regulators and transcription factors, therefore, TERRA mediated transcription regulations do not necessarily undergo PRC2 recruitment. Although TRF1 depletion convincingly increased SUZ12 level in certain genes (Figure 2 and 3), the large part of gene expression changes happened in genes without TERRA accumulation (Figure 7A). Therefore, most transcriptional changes are due to the systemic effect of TRF1 depletion rather than TERRA accumulation in such genes. Nevertheless, a significant portion of altered gene expression in shTRF1 cells coincide with TERRA bound genes (Figure 7A, 98/328 of down-regulated genes and 162/483 of up-regulated genes) and PRC2 (SUZ12) accumulation was clearly observed at pluripotency and differentiation genes. This study does not sufficiently support a model in Figure 8 as also claimed in the manuscript, but still can deliver important information of TRF1 functions on stem cells.

The reviewer has an interesting point. We would like to clarify that the CHIRT technique can only detect the RNA loops formed by direct RNA-DNA interaction. Therefore, a defined TERRA peak by CHIRT indicates the place in the genome where the RNA-DNA interaction is taking place. However, TERRA RNAs are very long and can spread way further than the point of interaction with DNA. In this scenario, it is possible that TERRA is recruiting PRC2 components in places non-coincident with TERRA peaks. That could explain in part that only a fraction of TERRA peaks correlates with SUZ12 peaks and that many SUZ12 peaks do not show TERRA peaks. We now explain this in the revised manuscript text (see subsection “Genomic TERRA binding correlates with genes that are differentially expressed in the absence of TRF1 and that have presence of SUZ12 and H3K27me3”, third paragraph).

In addition, and in agreement with the reviewer´s remark, part of SUZ12 binding to the genome could be independent on TERRA recruitment due to changes in the epigenomic status of the cell. Alternatively, this could be due to the changes on gene expression that we observe upon TRF1 depletion (see Figure 1). We discuss now this subject in the revised manuscript text (see the aforementioned paragraph).

3) H3K27me3 is an epigenetic mark tightly associated with gene suppression. However, this study demonstrated that a number of genes are upregulated with increased H3K27me3. Authors need to provide a sufficient explanation of these seemingly contradictory results.

The reviewer has an interesting point. Cluster 1 in Figure 4D shows the genes whose expression is upregulated upon TRF1 depletion and present an increase of H3K27me3 around their TSS. It is important to note that the increase in H3K27me3 mark is very low. Most importantly, analysis of these genes shows that almost 70% of them belong to the group of bivalent genes. It is described that bivalent genes show a lower presence of H3K27me3 in naïve state that in primed state (Marks et al., 2012). Our data support the idea that depletion of TRF1 induces a transition from naïve to primed-differentiated state. The increased signal of H3K27me3 in those bivalent genes upon TRF1 depletion further supports this model. We now discuss this in the revised manuscript text (subsection “Depletion of TRF1 induces recruitment of SUZ12 and H3K27me3 to genes de-regulated in the absence of TRF1”, second paragraph).

4) Physiological information is somewhat missing in this study. Authors can enhance their point on the TRF1 function by measuring TRF1 and TERRA level changes during early cell differentiation.

We have previously described that in vivo de-differentiation in mouse tissues as the consequence of in vivo reprogramming results in increased TRF1 levels in the de-differentiated tissue areas coincidental with expression of the Oct4 pluripotency factor (Marion et al., 2017). In turn, we previously showed that TRF1 levels are higher in the Inner Cell Mass (ICM) at the blastocyst stage in mice compared to differentiated mouse embryonic fibroblasts (MEFs) (see Figure 4C, D in Varela et al., 2011). In addition, we have previously shown that TRF1 is upregulated in stem cells compartments compared to the differentiated compartments in different adult tissues (Schneider et al., 2013). Together, these previous findings from our group clearly point to TRF1 regulation associated to differentiation state. Similarly, we have previously shown that TERRA is upregulated during induction of iPS cells (Marión et al., 2009).

Nevertheless, we have addressed this issue with an additional experiment in which we now demonstrate that TERRA and TRF1 levels are also downregulated when we induce differentiation of ES cells upon treatment with retinoic acid (see new Figure 5—figure supplement 2).

5) TRF1 is not a sole shelterin component regulating TERRA, previous studies show greater TERRA upregulation upon TRF2 depletion (Porro et al., 2014; Rossiello et al., 2017). Authors may make clear the insight on the TERRA mediated gene regulations with TRF2 depleted p53 -/- cells.

We thank the reviewer for this interesting suggestion. In the revised manuscript text, we now show that depletion of TRF2 by an shRNA in 2i-grown p53^-/-^iPS does not induce changes in TERRA expression (see new Figure 5—figure supplement 1B-C).

Reviewer #2:[…] The data are interesting as they might indeed explain why TRF1 is involved in pluripotency maintenance, yet the study remains largely descriptive/correlative. While the authors' model makes sense with the data, some validation experiments are necessary to validate the proposed mechanisms. Depleting TERRA in TRF1-depleted cells followed by RNA-Seq or RT-qPCR of selected genes would confirm of disprove the model. Anti-TERRA gapmers have been used successfully in mouse cells, thus this experimental plan should not pose major difficulties.

We thank the reviewer for the detailed review of our manuscript and for considering that “The data are interesting as they might indeed explain why TRF1 is involved in pluripotency maintenance”. The reviewer also has a number of insightful suggestions for new experimentation and revisions, which we have addressed in the revised manuscript. In particular, we have now included in the revised manuscript the following new sets of data:

As suggested by the reviewer, we have now downregulated TERRA levels in TRF1-depleted iPS (p53-null). Due to technical incompatibilities with the use of shRNAs and Gapmers, we used siRNAs to downregulate TRF1 levels and Gapmers to decrease TERRA expression. Thus, 2i-grown *p53^-/-^*iPS cells were co-transfected with a combination of two different siRNAs against TRF1 and a Gapmer against TERRA. In this experimental setting, we obtained up to 25-30% downregulation of TERRA levels in these cells by using Gapmers (see Figure 7—figure supplement 1A) and a moderate decrease in TRF1 levels (see Figure 7—figure supplement 1B). In spite of the technical difficulties of the experiment and the moderate downregulation of TRF1 and TERRA, we went ahead and performed RNA-seq of the samples. We observed that a number of genes that we had found altered when depleting TRF1 by shRNA were also regulated when depleting TRF1 with an siRNA (see Figure 7—figure supplement 1C), although the changes were modest, most likely owing to the modest downregulation of TRF1. Importantly, we now show that expression of some of these genes was not altered when downregulation of TERRA was performed at the same time that depletion of TRF1 (see Figure 7—figure supplement 1C), indicating that their expression change depends on TERRA upregulation. Interestingly, ENRICH analysis of the downregulated genes showed that they are targets of pluripotency factors and belong to the PluriNetWork pathway (see Figure 7—figure supplement 1D), while upregulated genes are targets of pluripotency factors, differentiation factors and Polycomb, and belong to differentiation pathways (see Figure 7—figure supplement 1E). Therefore, these results are in agreement with our model, in which the upregulation of TERRA levels is required for the changes in gene expression produced upon TRF1 depletion.

TERRA upregulation in TRF1-depleted cells is somehow surprising as the de Lange group showed no changes in TERRA levels in TRF1-deleted MEFs. Could this be linked to the p53 status of the used cells? That specific paper needs to be referenced as it was the first one describing a TRF1 KO mouse.

The reviewer has a good point. In the de Lange paper the experiment is done in immortalized MEFs, while we use either p53 KO iPS cells or keratinocytes. These differences in cell type and immortalization conditions could explain the contradictory results. In any case, we clearly show here a dramatic increase in TERRA in two different mouse cells types (iPS cells and keratinocytes) upon TRF1 depletion (see new Figure 5). Moreover, several works from other groups (Porro et al., 2014; Sadhukan et al., 2018, Zeng et al., 2017) have shown TERRA increase upon TRF1 depletion in human cells. Have now cite the De Lange paper and the three other papers in the revised manuscript text, as well as discuss this potential discrepancy (see subsection “Depletion of TRF1 in 2i-grown iPS cells induces the up-regulation of TERRA RNAs expression”, first paragraph).

Moreover, the Northern blots shown in Figure 5B-D are not satisfactory. TERRA pattern is different in each panel, there is no TERRA signal in panel C for MEFs, while mouse TERRA is definitely abundant and easily detectable. Similarly, there is no signal for TERRA in wt cells in panel D. Why does the 18S signal fade away till disappearing throughout samples in panel C?It would be important to test whether the increased in TERRA depends on transcription.

The reviewer has a good point. We have now repeated the Northern blot shown in former Figure 5C using new MEFs and a loading control. As expected, we see higher levels of TERRA in iPS when compared with MEFs, and a further increase in TERRA level upon TRF1 depletion (see new Figure 5C). Regarding WT cells for panel D, the levels of TERRA expression for WT keratinocytes are very low and then difficult to see. We have now included images for an independent experiment when, upon higher exposure, levels of TERRA levels in WT keratiyocytes can be better observed. Again, the results confirm that depletion of TRF1 highly increases the levels of TERRA (see Author response image 1). Regarding the reviewer concern of whether increase in TERRA levels upon TRF1 depletion depends on transcription, our CHIRT results indicate that there is an important presence of TERRA bound to the subtelomere of the chromosome 18, a locus previously described by us as a *bona fide* TERRA locus (Figure 6C), and that this binding is increased upon TRF1 depletion (Figure 7J), supporting the idea that more TERRA transcription is indeed occurring when TRF1 is depleted.

Reviewer #3:In this manuscript, the authors (Marion, et al.) investigate the function of TRF1 in pluripotent cells and demonstrate that the TRF1 removal leads to the changes of gene expression that are related to the pluripotency. They also found that the enrichment of SUZ12 and H3K9me3 were increased in genes that are involved in the development and cell differentiation. In the end, they demonstrated that a long non-coding RNA, TERRA, is induced upon TRF1 depletion and the increase of TERRA binding to genome correlates the patterns of SUZ12 and H3K9me3 enrichment. The authors propose a model that TERRA mediates the recruitment of PRC2 complex in TRF1 depleted cells. Their findings are exciting and provide a mechanism by which TRF1 not only functions at telomeres but also is essential for the epigenetic regulation in ES cells. However, there are a few concerns that should be addressed that could make the manuscript more convincing and more interesting for the readership of eLife.1) There is no direct evidence showing the TERRA mediates the increased recruitment of SUZ12 to those genomic loci after TRF depletion in this study. Does TRF1 removal increase the interaction between PRC2 and TERRA in vivo? This is an important question that could greatly support the author's claim. RNA immunoprecipitation (such as UV-RIP) for PRC2 complex could be done to address this question.

As suggested by the reviewer, in the revised manuscript we now include a RIP assay that demonstrates a higher binding of SUZ12 protein to TERRA upon TRF1 depletion (see new Figure 8).

Figure 7D and 7E show that the binding sites of SUZ12 and TERRA are nearby but not at the same sites. If TERRA mediates the PRC2 recruitment, why SUZ12 and TERRA bind at different sites?

We thank the reviewer for the observation. As The CHIRT technique can only detect the RNA loops formed by direct RNA-DNA interaction. Therefore, a defined TERRA peak by CHIRT indicates the place in the genome where the RNA-DNA interaction is taking place. However, TERRA RNAs are very long and can spread way farther than the point of interaction with DNA. In this scenario, it is possible that TERRA is recruiting PRC2 components in places non coincident with TERRA peaks. That could explain why binding sites of SUZ12 and TERRA are nearby but not at the same sites. We now discuss this in the revised manuscript text (subsection “Genomic TERRA binding correlates with genes that are differentially expressed in the absence of TRF1 and that have presence of SUZ12 and H3K27me3”, third paragraph).

2) I couldn't find anywhere that the authors mentioned the biological replicates for ChIP-seq data. If not, it is crucial to have at least two biological replicates of ChIP-seq or ChIP-qPCR to confirm the increase of SUZ12 and H3K27me3 enrichment at several genomic loci after TRF1 depletion.

In our current study we have used two biological replicates. In Figure 2—figure supplement 1 and Figure 2—figure supplement 1 and Figure 2—figure supplement 4, we now confirm the increase of SUZ12 and H3K27me3 at several genomic loci in both replicates, as well as the binding of TRF1 to extratelomeric sites (see new Figure 2—figure supplement 1 and Figure 2—figure supplement 4). In particular, we found that heatmaps of the two biological replicates of SUZ12 ChIP-seq, with reads plotted around the peaks for both replicas, show the same pattern (see new Figure 2—figure supplement 1A). Importantly, the analysis of the genomic regions where SUZ12 signal was increased was highly similar (see new Figure 2—figure supplement 1A). Thus, we observed that all the clusters of these heatmaps included regions when SUZ12 increases upon TRF1 depletion in both replicas (see new Figure 2—figure supplement 1A). Importantly, we show that 99% of the genes that show increased levels of SUZ12 upon TRF1 depletion in the first experiment also show higher levels of SUZ12 in the second replica, confirming the high similarity between both ChIP-seq and the recruitment of SUZ12 to the described genomic sites (see new Figure 2—figure supplement 1C). We also find that heatmaps of the two biological replicates of H3K27me3 ChIP-seq, with reads plotted around the peaks for both replicas, show the same pattern (see new Figure 2—figure supplement 1D). We observed that cluster 3 and 7 of these heatmaps include regions when H3K27me3 increases upon TRF1 depletion in both replicas. Importantly, we show that 90% of the genes that show increased levels of H3K27me3 upon TRF1 depletion in the first experiment also show higher levels of H3K27me3 in the second replica (see new Figure 2—figure supplement 1F), confirming the high similarity between both ChIP-seq. In the case of the two biological replicas of TRF1 ChIP-seqs, we find that the number of genomics sites bound by TRF1 is very low in both replicas (see new Figure 2—figure supplement 4), suggesting that the vast epigenetic changes observed here as the consequence of TRF1 depletion and which involve re-localization of PRC2 to many genomic sites, are not likely to be explained by binding of TRF1 to these few extra-telomeric sites. Also, in both replicas the binding of TRF1 to the genome is mainly associated to extratelomeric sequences (see new Figure 2—figure supplement 4).

3) The title seems to contradict the author's results in the manuscript. TERRA recruits more PRC2 complex to the genomic loci in the absence of TRF1. So it means that TERRA can regulate the transcriptional landscape of pluripotent cells through a TRF1-independent manner (not dependent). In the fourth paragraph of the Discussion, the authors also state that TERRA can also bind to PRC2 sites in the genome that are dependent on TRF1. If TERRA binding to PRC2 sites is in a TRF1-dependent manner, the results should show the decrease of TERRA binding to PRC2 sites upon TRF depletion.

We would like to clarify this point. Since PRC2 is recruited to the genome by

TERRA expression regulation, and TERRA expression is regulated by TRF1 we consider adequate to state that PRC2 recruitment and TERRA expression is dependent on TRF1. By saying that is dependent on TRF1 we don´t imply that is dependent on the presence of TRF1, but on the regulation, function and expression of TRF1 protein.

[Editors' note: further revisions were requested prior to acceptance, as described below.]

The manuscript has been improved but there are some remaining, significant issues that need to be addressed before acceptance, as outlined below:Reviewer #2:The paper has now improved, yet some main issues remain.1) The TERRA depletion experiment is not satisfactory according to me. First of all, a 10/20% depletion efficiency is essentially no depletion. With such depletion efficiencies I would argue that any observed cellular responses derive from introducing a C-rich telomeric RNA sequence in cells, rather than from depletion of TERRA. Additionally, measuring TERRA depletion using FISH is not ideal as the signal might diminish at least in part because the C-rich gapmer is preventing the FISH probe from binding, possibly in absence of TERRA degradation in cells. A Northern blot should be performed. Also, how does the TERRA signal look like? There is no image shown, as to give a sense of how this minor depletion can be visualized. A complementation experiment where TERRA is ectopically re-expressed is formally impossible using the telomeric C-rich gapmers, as the telomeric TERRA sequence cannot be changed. Nevertheless, the authors could also consider to deplete TERRA using one or more gamers targeting unique sequences from the subtelomere of chromosome 18, which is the TERRA locus according to results from the Blasco lab. In this case, a complementation experiment could be performed by expressing ectopic telomeric RNA, which should localize in trans to all telomeres, again according to what published by the Blasco lab. Issues stem also from the fact that siRNA-mediated depletion of TRF1 is very poor (at least at the mRNA levels; I am not sure I understand why the authors did not perform a Western blot). Why are shRNA and gapmers 'incompatible'? Overall, as they stand now, I do not think that those experiments should be published.

We understand the reviewer´s concern regarding the partial depletion efficiency of TERRA in our experiments, and following his/her suggestion we are willing to remove these results from the revised manuscript. That said, we would like to explain all the attempts that we carried out to downregulate TERRA and the technical difficulties that we encountered. The reviewer asks about how TERRA signal looks like in the RNA FISH experiments. We now provide him/her with representative images of these experiments (see Author response image 2). Regarding his concern about measuring TERRA depletion by FISH, we will like to mention that we have measured TERRA depletion also by Northern Blot and confirmed the results shown by RNA FISH, with a modest TERRA downregulation (see Author response image 2).

**Author response image 2. respfig2:** 

The co-silencing of TRF1 and TERRA in iPS cells has proven to be a technically difficult experiment. Since TERRA is upregulated upon TRF1 depletion, downregulation of TERRA has to be achieved since the very first moment TRF1 is silenced, to avoid the activation of downstream consequences of TERRA expression. This timing requirement adds an extra difficulty to the experimental approach. We first tried several times to induce the co-silencing using shRNA against TRF1 and TERRA GAPMERS. As mentioned, TERRA should be silenced since the first time of shRNA infection to the last one, and maintained during the posterior antibiotic selection. During the infection process with shRNAs, cells are infected twice a day, so the medium is changed. This makes difficult to maintain the constant presence and facilitate the transfection of TERRA GAPMERS in a cell type, iPS cells, when transfection efficiency is already low. Also, after the infection process, the constant presence of antibiotic to select for infected cells interferes with the transfection of TERRA GAPMERS. We performed this experiment twice, with variations in the timing of TERRA GAPMERS transfection, and in any case we could not detect even a modest reduction of TERRA levels. It is worth to mention that measurement of TERRA by RNA FISH did not show any TERRA downregulation, indicating that introduction of C-rich GAPMERS is not preventing the FISH probe from binding.

Then, and to avoid the infections required for the use of shRNAs, we decided to downregulate TRF1 using siRNA in a double transfection experiment. With this approach, we have also performed several attempts to optimize the proper timings of infection to achieve the best TRF1 and TERRA downregulation. However, transfection of iPS cells is an inefficient process, and the chances that the same cell is transfected with both the siRNA and GAPMERs is low. Still, with this approach we have been able to obtain the moderate decrease (10-20%) shown in the manuscript. We believe that by using GAPMERs against the subtelomere of chromosome 18 we will encounter the same technical problems and we will not improve TERRA downregulation.

We understand that this level of downregulation is low and not optimal and thus, as suggested by the reviewer, we have removed this experiment from the revised manuscript. In any case, we believe that the main message of the paper remains the same, and represents a significant advance in our understanding of the role of TRF1 in pluripotency.

2) I still think that the quality of the Northern blots shown in Figure 5 is quite low. For panel D (which did not change compared to the last version), I suggest using the images shown in Author response image 1, they are definitely better.

We thank the reviewer for his/her suggestion. Following such suggestion, we have changed the representative image of Figure 5D in the revised manuscript figures (see new Figure 5D).

3) The authors now back up their finding in mouse cells by citing several reports showing TERRA up-regulation in TRF1-depleted human cells of various type. This is ok and possibly interesting in terms of evolutionary conservation of this mechanism. Unfortunately, the authors forgot to cite their own paper appeared in Nat Cell Biol in 2008 (Schoeftner and Blasco, 2008), where siRNA-mediated depletion of TRF1 in C2C12 mouse cells was shown to decrease TERRA levels, and TRF1 was thus presented as a negative regulator of TERRA transcription through its interaction with RNA polymerase II. This should also be explained.

The reviewer is right and this paper should have cited and discussed. In the revised manuscript we have now included the citation and discussion about these results (see subsection “Depletion of TRF1 in 2i-grown iPS cells induces the up-regulation of TERRA RNAs expression”, first paragraph).

Reviewer #3:The revised version of the manuscript provides some additional work such as RNA-seq and RNA-IP experiments to strengthen the authors' idea that TRF1 regulating gene expression in pluripotent cells is mediated by TERRA and SUZ12 recruitment to the genome. They found that the interaction of TERRA and SUZ12 was increased after TRF1 depletion. They also used siRNA and LNA Gapmer to knockdown TRF1 and TERRA in 2i-grown iPS cells and investigated the transcriptome changes between TRF1-alone depletion and TRF1/TERRA depletion. They show that the expression of some genes, which were altered in TRF1-alone depletion, was not altered in TRF1/TERRA depletion, suggesting that the upregulation of TERRA levels is required for the changes in gene expression upon TRF1 depletion. However, the authors didn't compare the results with the gene targets for the increase of TERRA binding and SUZ12 binding after TRF1 depletion. Therefore, it is not clear whether the recruitment of TERRA and SUZ12 is required for the regulation of those genes. This question needs to be addressed before publication.1) In Figure 7—figure supplement 1C, the authors show that the expression of some genes that were not altered when downregulation of TERRA after TRF1 depletion (such as Lef1 and Stk39), indicating that TRF1 regulates the expression of these genes by regulating TERRA levels. Do these genes have TERRA binding sites and SUZ12 binding sites? And do they correlate with the increase of TERRA binding and SUZ12 binding after TRF1 depletion? These questions are important to rule out whether TERRA regulates the gene expression is mediated by recruiting SUZ12 or other mechanisms. In Figure 7J, the authors also show that up or down-regulated genes after TRF1 depletion have TERRA binding sites nearby. Does the expression of those genes change in both TRF1 and TERRA depleted cells? Because I don't see any genes are presented in both Figure 7—figure supplement 1C and Figure 7J, the connection between TRF1 depletion, the increase of TERRA binding sites, and the increased of SUZ12 binding sites is weak. It would be more convincing that the authors show the gene expression of some specific genes (up or down-regulated in shTRF1 cells having TERRA and SUZ12 binding sites), which are controlled by TERRA and SUZ12 in a TRF1 dependent manner (TRF1 expression). In other words, the authors should find some common genes in Figure 7J and Figure 7—figure supplement 1C. The authors could check TERRA CHIRT qPCR for those genes found in Figure 7-supplement 1C using shTRF1 cells.

In this figure we show the 46 genes whose expression is altered in a similar way when downregulating TRF1 by siRNA and depleting TRF1 by shRNA (as measured by RNA-seq). Regarding his/her question about whether these genes show TERRA and SUZ12 binding sites and increased binding of SUZ12 upon TRF1 depletion, 39 genes show binding for SUZ12 (as obtained from heatmaps in Figures 3D and 4D), and all of them show increased binding of SUZ12 upon TRF1 depletion. In terms of TERRA, 11 of these genes show binding of TERRA (as obtained from heatmaps in Figure 7B and 7C) and 9 of them present increased binding of SUZ12 upon TRF1 depletion. Regarding his/her question about whether expression of genes shown in Figure 7J change in both TRF1 and TERRA depleted cells the answer is that none of them show any change, probably due to the low TERRA downregulation obtained in this experiment. Owing to the poor efficiency in TERRA downregulation, and as suggested by reviewer #2, we are removing the results shown in Figure 7—figure supplement 1C from the manuscript.

2) The CHIRT-seq technique can detect RNA interacting chromatin that could be either mediated by protein complexes or through direct RNA/DNA interaction. The sentence claiming that "The CHIRT technique can only detect peaks where RNA-DNA interaction occurs." in the main text is incorrect and misleading. CHIRT-seq (Chu et al., 2017) utilizes glutaraldehyde to fix cells, and this reagent will crosslink the protein-protein complex, protein-RNA, protein-DNA complex, and RNA-protein-DNA complex. Other methods such as ChIRP-seq (Chu et al., 2012) developed by Howard Chang's laboratory and CHART-seq (Simon et al., 2013) developed by Robert Kingston's laboratory also use aldehyde-based chemicals to crosslink RNA-chromatin complex. The major purpose of these techniques is to detect RNA associated complexes that are made of RNA, proteins, and DNA. The authors need a better explanation of why SUZ12 and TERRA have different binding sites. One possibility is that ChIP and CHIRT use physical sonication to shear chromatin, and thus the long-range interaction mediated by long non-coding RNA could be disrupted after sonication. Other mechanisms that may lead to this result should also be discussed in the Discussion section.

We thank the reviewer for this correction. In the revised manuscript we have now removed that sentence and include alternative explanations to explain why SUZ12 and TERRA have different binding sites (see subsection “Genomic TERRA binding correlates with genes that are differentially expressed 486 in the absence of TRF1 and that have presence of SUZ12 and H3K27me3”, third paragraph).